

# Deformation of intrasalt competent layers in different modes of salt tectonics

Mark G. Rowan[1], Janos L. Urai[2], J. Carl Fiduk[3], Peter A. Kukla[4]

[1]Rowan Consulting, Inc., Boulder, CO 80302, USA
5 [2]Institute for Structural Geology, Tectonics and Geomechanics, RWTH Aachen University, 52056 Aachen, Germany
[3]Fiduk Consulting LLC, Houston, TX 77063, USA
[4]Geological Institute, Energy and Mineral Resources, RWTH Aachen University, 52056 Aachen, Germany

*Correspondence to*: Mark G. Rowan (mgrowan@frii.com)

10 **Abstract.** Layered evaporite sequences (LES) comprise interbedded weak layers (halite and, commonly, bittern salts) and strong layers (anhydrite and usually non-evaporite rocks such as carbonates and siliciclastics). This results in a strong rheological stratification, with a range of effective viscosity up to a factor of $10^5$. We focus here on the deformation of competent intrasalt beds in different modes of salt tectonics using a combination of conceptual, numerical and analog models, and seismic data. In bedding-paralell extension, boudinage of the strong layers forms ruptured stringers, within a 15 halite matrix, that become increasingly isolated with increasing strain. In bedding-parallel shortening, competent layers tend to maintain coherency while forming harmonic, disharmonic, and polyharmonic folds, with the rheological stratification leading to buckling and fold growth by bedding-parallel shear. In differential loading, extension and the resultant stringers dominate beneath suprasalt depocenters while folded competent beds characterize salt pillows. Finally, in tall passive diapirs, stringers generated by intrasalt extension are rotated to near vertical in tectonic melanges during upward flow of salt. In all 20 cases, strong layers are progressively removed from areas of salt thinning and increasingly disrupted and folded in areas of salt growth as deformation intensifies. The varying styles of intrasalt deformation impact seismic imaging of LES and associated interpretations. Ruptured stringers are often visible where they have low dips, as in slightly extended salt layers or beneath depocenters, but are usually not imaged in tall passive diapirs due to steep dips. In contrast, areas of slightly to moderately shortened salt typically have well imaged, mostly continuous intrasalt reflectors, although seismic coherency 25 decreases as deformation intensifies. Similarly, wells are most likely to penetrate strong layers in contractional structures and salt pillows, less likely in extended salt because they might drill between stringers, and unlikely in tall passive diapirs because the stringers are near-vertical. Thus, both seismic and well data may be interpreted to suggest that diapirs and other areas of more intense intrasalt deformation are more halite rich than is actually the case.



## 1 Introduction

The "salt" of ancient global salt basins never comprises solely halite (or rocksalt). Instead, the stratigraphic unit that forms salt diapirs or serves as a salt detachment is better termed a layered evaporite sequence (LES) consisting of interbedded halite, other evaporites, and usually non-evaporite rocks. These sequences were deposited in isolated basins in arid climates, with the surface of the evaporitic brine ranging from very shallow to 1.5 km or more beneath global sea level (see Warren, 2016).

The most common non-halite evaporite mineral is gypsum/anhydrite (used interchangeably here) because, ignoring minor carbonate evaporites, it precipitates prior to halite as a body of water is evaporated (e.g., Warren, 2016). Anhydrite may occur in layers distributed throughout the LES but is often concentrated near the base and sometimes near the top due to generally less evaporative conditions at those times (Fig. 1). Bittern salts such as sylvite and carnallite, which require a large degree of evaporation, are less common but occur in many salt basins.

The South Atlantic (Fig. 1a) and Gulf of Mexico LES comprise almost exclusively evaporites (e.g., Salvador, 1991; Davison, 2007; Gamboa et al., 2008). Existing data show that any non-evaporite interbeds are rare and confined to the proximal and/or lateral edges of the salt basins. These basins are anomalies, however, in that the vast majority of salt basins include often significant proportions of non-evaporite rocks (Fig. 1b-e), with marine or lacustrine carbonates and mudstones being the most common. Examples include LES in such disparate salt basins as the Barents Sea, the Permian Basin of Europe, the Pyrenees and Betics of Spain, the Levant Basin in the eastern Mediterranean, the Red Sea and Gulf of Suez, the Oman salt basins, the Precaspian Basin, the Kuqa Basin of China, the Flinders Ranges of Australia, La Popa Basin in Mexico, the Paradox Basin of SW USA, and the salt basins of Nova Scotia (e.g., Dalgarno and Johnson, 1968; Hite and Buckner, 1981; Dekker, 1985; Wade and MacLean, 1990; Lawton et al., 2001; Peters et al., 2003; Volozh et al., 2003; Chen et al., 2004; Bosworth et al., 2005; Geluk, 2007a; Feng et al., 2016; Cámara and Flinch, 2017; Flinch and Soto, 2017; Rowan and Lindsø, 2017). Siliciclastic layers can also include siltstones, sandstones, and even conglomerates (e.g., Paradox Basin, Rasmussen, 2014). Moreover, LES can contain extrusive and intrusive igneous rocks that were emplaced into the depositional salt basin (Fig. 1b), for example in La Popa Basin, the Spanish Pyrenees, the Great Kavir Basin of Iran, and the South Oman Salt Basin (Jackson et al., 1990; Garrison and McMillan, 1999; Peters et al., 2001; Cámara and Flinch, 2017). In this paper, we use the terms *salt* and *LES* to refer to the entire stratigraphic package of evaporite and non-evaporite rocks; halite and other evaporite minerals are referred to by name.

The internal deformation of salt bodies, whether diapiric or not, can be observed in mines, in outcrop, and sometimes on modern seismic data (Fig. 2). One problem with outcrop data is that most surface exposures of salt comprise caprock, where dissolution of the halite and bittern salts has modified the original internal structure (Fig. 2a). Exposures of halite are rare, occurring in places such as Mt. Sedom in Israel (Fig. 2b), the Zagros Mountains of Iran, and the Kuqa Basin of China (e.g., Talbot, 1998; Li et al., 2014; Alsop et al., 2015). Mines that target halite or bittern salts provide data beneath the caprock but are limited by the available network of mine workings. Detailed maps and cross sections from mines in, among other places,

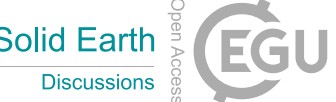

the Southern Permian Basin of northern Europe (e.g., Richter-Bernburg, 1980; Schachl, 1987; Burliga, 1996; Bornemann et al., 2003) depict the complex large-scale geometries (Fig. 2c), and cut faces often show beautiful examples of small- to mesoscale deformation (Fig. 2d). Modern seismic data, especially 3-D depth-migrated data (e.g., Gamboa et al., 2008; Van Gent et al., 2011; Fiduk and Rowan, 2012; Cartwright et al., 2012; Strozyk et al., 2012, 2014; Gvirtzman et al., 2013;

Jackson et al., 2014, 2015; Raith et al., 2016; Feng and Reshef, 2016; Feng et al., 2016), can reveal the full three-dimensional geometry but lack the resolution to image the small-scale structure, especially in mobile lithologies (Fig. 2e).

Studies of observed intrasalt deformation can be supplemented by the results of both analog and numerical models (e.g., Koyi, 2001; Cartwright et al., 2012; Albertz and Ings, 2012; Li et al., 2012a; Dooley et al., 2015). These models, although limited by the materials or properties used and the imposed boundary conditions, can demonstrate the effects of varying

individual parameters in simplified setups. As such, they provide significant help in evaluating and understanding natural systems.

The most common features within deformed salt bodies are folds, with varying styles and orientations, and isolated blocks of more competent rocks known as stringers. Although there are numerous published examples of intrasalt deformation, there has been little attempt to distinguish between the processes and geometries characteristic of different

styles of salt tectonics. In other words, do structures such as salt-cored anticlines, passive diapirs, and salt glaciers display different styles of internal deformation? Jackson and Hudec (2017) discuss intrasalt deformation in autochthonous structures, steep diapirs, and allochthonous sheets, but their primary focus is on folds associated with flow of viscous halite. Relatively minor attention is paid to the deformation of competent layers.

In this paper, we focus on the more competent non-halite/bittern components of the LES, investigating the internal

deformation in different scenarios. Specifically, we discuss the intrasalt processes and resulting geometries formed during extension, contraction, differential loading, and passive diapirism (including allochthonous sheets). Our primary emphasis is on the role of competency contrasts and consequent differences in the mechanics of deformation, not on any associated density contrasts. Moreover, we address only those non-halite/bittern rocks that formed part of the depositional LES; we do not include younger material that may have been entrained into salt at a later time, such as post-salt igneous intrusions or

encased minibasins and salt-sheet carapace along sutures in salt canopies. Nor do we address the deformation commonly observed in outcrop due to the modification of original fabrics by dissolution. Our approach is to some degree conceptual, but is founded upon observations of real structures, the results of analog and numerical models, and a thorough understanding of the underlying mechanics. Note that we discuss only those non-halite layers that are thick and strong enough to form independent mechanical units; very thin beds of anhydrite or shale tend to behave mechanically with the

encasing halite, as seen within mines in the Cardona diapir of Spain or the Wieliczka diapir of Poland.

This analysis of intrasalt deformation is relevant to anyone concerned with the interiors of salt bodies. This includes, of course, those extracting any resources (halite, bittern salts, metals, or hydrocarbons) from within salt, those drilling through salt in the search for hydrocarbons, and those looking to store materials (hydrocarbons, $CO_2$, nuclear waste) within diapirs. The internal geometries impact the building of velocity models for depth migrations, the transmission of acoustic energy




through salt, and the interpretation of the resulting images (e.g., Jones and Davison, 2014). Potential drilling problems include creep of the more mobile halite and especially bittern salts, washouts and loss of circulation, and kicks due to overpressure in stringers or fluid pockets (e.g., Kukla et al., 2011; Strozyk, 2017). A better understanding of the internal structure in different settings will help to reduce risk.

In the following sections, we first summarize relevant aspects of the rheology, then address intrasalt deformation in simple extension, contraction, differential loading, and passive diapirism. In each case, we illustrate the concepts with a combination of schematic diagrams, seismic images, and numerical/analog models. We then discuss complications such as the definition of the top and base of the LES, the roles of competent-layer properties such as thickness and density, and the mechanical stratigraphy of LES multilayers. Furthermore, because many salt structures experience combinations of the

simple end-member processes, either simultaneously or consecutively, we describe and evaluate several real examples with histories that vary either spatially or temporally. Despite the possible complexities, however, we argue that there are fundamental differences in style that can be used to help distinguish between different modes of salt tectonics. Finally, we elaborate on some of the practical implications for seismic imaging, seismic interpretation, and drilling into and through salt.

## 2 Rheology

The rheology of an LES is strongly layered, with contrasts in effective viscosity of up to five orders of magnitude. Here, we summarize the rheology of halite, bittern salts, anhydrite, and non-evaporite layers.

### 2.1 Halite

Halite, or rocksalt, is weak and ductile under most conditions of geologic deformation; natural fractures only form either close to the surface, as in the Mt Sedom Diapir of Israel, or when penetrated by fluids with pressures higher than the

minimum principle stress in the salt. Experimentally derived flow laws have been extrapolated to the much lower strain rates of nature based on studies of the microscale deformation mechanisms operating under these conditions (see reviews by Carter and Hansen, 1983; Albrecht et al., 1990, Urai et al., 2008). Microstructural studies of naturally deformed halite show that dislocation glide and dislocation creep processes, solution-precipitation creep, and water-assisted dynamic recrystallization are all of major importance (Urai et al., 1987; Spiers and Carter 1998; Schléder and Urai, 2005, 2007), with

the relative importance varying as a function of grain size, impurity content, stress path, temperature, and fluid chemistry. During fluid-assisted dynamic recrystallization of rocksalt in nature (water content >10 ppm), the grain size adjusts itself so that the material deforms close to the boundary between the dislocation-creep and pressure-solution-creep fields (Fig. 3a; Spiers & Carter 1998). Power-law flow is thus proposed to be a good representation of this behavior. Differential stress in naturally deforming rocksalt (Fig. 3a) generally ranges between 0.5 and 2 MPa but can be as high as 5 MPa at shallow levels

(Spiers and Carter 1998; Schléder and Urai 2005, 2007); grain size is typically around 10 mm in diapiric halite and effective viscosities at these differential stresses are between $10^{18}$ and $10^{19}$ Pa s.



### 2.2 Other evaporites

Bittern salts such as carnallite and bischofite deform in a manner similar to that of halite but are significantly weaker than rocksalt (Urai, 1983, 1985; Urai et al., 1986). In general, bittern salts have effective viscosities 100 to 1000 times lower than that of halite and thus flow 100 to 1000 times faster than halite under a given differential stress (Fig. 3a). When present, they

form the weakest layers of the LES.

    The rheology of anhydrite in the subsurface is not well constrained by extrapolation from laboratory experiments, probably because solution-deposition processes are much more important in nature. In shortening, anhydrite behaves as a viscous material, but the shape of concentrically folded anhydrite layers within halite indicates an effective viscosity 10-100 times that of rocksalt (Fig. 3a; Schmalholz and Urai, 2014). In layer-parallel extension, however, anhydrite behaves in a

brittle manner, readily deforming by boudinage with variable degrees of pinch-and-swell structures prior to eventual breakup into stringers. High pore pressures sustained by the surrounding impermeable rocksalt may impact how deformation is accommodated (Van Gent et al., 2011), and anhydrite weakens markedly at higher temperatures (Jackson and Hudec, 2017).

### 2.3 Non-evaporite layers

Whereas there is strong evidence for anhydrite being viscous under layer-parallel shortening and geological strain rates,

other strong interbeds such as carbonates, mudstones, and sandstones are materials deforming according to frictional-plastic constitutive laws. There can be a large range of strength depending on the lithology, the burial and diagenetic history and consequent degree of cementation, and the amount of overpressure sealed by of the surrounding halite. Nevertheless, these interbeds are typically considerably stronger than the rocksalt (Fig. 3b). In addition, frictional-plastic materials are significantly weaker during extension than they are during contraction, failing earlier (and again, in a brittle manner) during

extension (Fig. 3b).

### 2.4 LES rheology

As a package, the bulk rheology of an LES will vary between the Voigt and Reuss bounds (homogeneous strain and homogeneous stress, respectively; see Hill, 1952) and is strongly anisotropic: it is weakest in bedding-parallel simple shear and strongest in bedding-parallel horizontal shortening. However, these end-members are rarely applicable to the complex

kinematics and geometries of salt deformation, leading to folding, boudinage of the competent layers, and bedding parallel shear that enhances multilayer folding. At high strains, the LES forms a tectonic melange which is less anisotropic and stronger than pure rocksalt.





## 3 Modes of salt tectonics

### 3.1 Extension

Competent layers encased within a weak, ductile matrix undergo boudinage during layer-parallel extension (e.g., Ramberg, 1955; Ramsay, 1967; Goscombe et al., 2004). With enough extension, the strong layers separate into distinct boudins, with

the ductile material flowing into the space created. In metamorphic rocks, length-to-thickness aspect ratios typically range from 2 to 4 (Fossen, 2010).

When formed by layer-parallel extension in salt, the boudins are termed stringers (Mattes and Conway Morris, 1990). During thin-skinned extension, the overburden extends by the development of normal faults that may have a preferred dip direction or may form conjugate sets. The faults detach within the salt, presumably in zones of concentrated shear, while

intrasalt strong layers form boudins that are distributed throughout the region where suprasalt extension occurs (Fig. 4b). If extension is thick-skinned, boudinage is likely to be concentrated in areas of drape folding above presalt faults (Fig. 4c). Interestingly, intrasalt stringers have significantly larger length to thickness ratios compared to those in metamorphic rocks (Burliga, 1996; Van Gent et al., 2011).

An analog model of gravity-driven thin-skinned extension shows some characteristics that may occur in nature. Intrasalt

strong layers are extended three to four times as much as the cover, with the amount of stretching increasing downward (Fig. 5a). This is due to the significant thinning and attenuation of the weak and ductile model material, with bulk basinward flow during translation of the overburden – a combination of pure shear and simple shear (Brun and Mauduit, 2009). Although this internal strain variation cannot be seen in the tectonically-driven thick-skinned example of Figure 5d, the intrasalt strong layer is more extended than the cover (where the extension is off the end of the line) or the presalt section. With increasing

extension, the cusps in the top salt created by suprasalt normal faults (Fig. 4b) evolve into isolated, triangular remnants termed salt rollers. These have more complex internal deformation (Brun and Mauduit, 2009), as recorded by the internal layering of the Santos Basin LES within a large salt roller (Fig. 5c).

The models of Figure 4 are plane strain, but if movement of the salt is divergent, more complex patterns arise (Abe et al., 2013). This might happen above a radially spreading allochthonous salt canopy (Rowan et al., 1999) or deltaic lobe (Gaullier

and Vendeville, 2005). Alternatively, drape folding from a common footwall to two presalt faults with different trends can lead to multi-directional extension (see Dooley et al., 2004), and crustal doming may cause outwardly directed gravity gliding and the development of a polygonal pattern of extension, as suggested for the pods and interpods of the Central North Sea (Karlo et al., 2014). Whatever the drive, non-uniaxial extension leads to tablet boudinage in which intrasalt strong layers are broken in multiple directions (e.g., Wegmann, 1932; Zulauf et al., 2011).

### 3.2 Contraction

The dominant structural style of contraction above a salt layer is that of detachment folds cored by the mobile salt. During early stages of shortening, the area in the core increases as the fold grows (Dahlstrom, 1990; Stewart, 1999); this is balanced





in a two-dimensional analysis by sinking of synclines, with underlying salt flowing laterally into the anticlines (Fig. 6). Because rocks are generally stronger in shortening than in extension, the competent intrasalt layers tend to remain coherent and unbroken, at least at relatively low strain. When the strong layers are continuous, again in a plane-strain model, the intervening areas of viscous salt maintain their area. Internal strong layers would thus seem to have progressively less length

downward because the uppermost part of the LES experiences similar amounts of shortening to the cover while the lowermost part is almost undeformed (Fig. 6b). In order to maintain bed length, therefore, the competent layers form fold trains or thrusts with increasing amounts of shortening toward the base of the salt (Fig. 6c). The deformation is commonly disharmonic or polyharmonic, with fold wavelengths dependent on layer thickness and strength, and folding is accommodated primarily by layer-parallel slip. As contractional strain increases over time, the strong layers may get

disrupted, especially beneath synclines and in the cores of anticlines, potentially leading to mixing of different halite-rich portions of the LES and thus more complex deformation.

    Experimental models with strong intra-polymer layers display some of these expected features (Fig. 7a, b). The layers generally form complex folds that are generally tight to isoclinal, range from upright to recumbent, and are commonly ptygmatic. They may verge in either direction due to overall flow of ductile material from beneath synclines into the cores of

adjacent anticlines (Fig. 7a). The anticlinal cores are where the strong layers are most likely to be ruptured due to more intense deformation. When there are two or more competent layers, the deformation is disharmonic and the deeper layer experiences more shortening (Fig. 7b).

    Seismic images may or may not match the analog models. In the contractional domain of the Santos Basin, for example, there is indeed more contraction at deeper levels of the LES than at shallower levels, which fold concordantly with the

overburden (Fig. 7e). In the Levant Basin, however, shallower strong layers appear to have greater amounts of shortening (Fig. 7d). This is probably due to the dominance of Couette flow (simple shear) in the Messinian evaporites (see Fig. 8a), as recorded by fluid escape pipes within the salt and its overburden (Cartwright et al., 2018).

    Although Couette flow is nicely illustrated in the Levant Basin, other shear profiles are possible (Fig. 8). These may include intrasalt basal shear, intrasalt basal shear with reverse Couette flow, and asymmetric Poiseuille flow. Intrasalt basal

shear seems to be important in the Santos Basin, with the intrasalt beds shortened above the unaffected presalt section, so why is this different from what is observed in the Levant Basin? The key may lie in the larger-scale geometry of the LES, and thus an evolving internal shear profile over time. When the salt layer is broadly tabular, with roughly parallel top and base (i.e., early in the deformation, as in the Levant Basin), Coutte flow may dominate (Fig. 9a). As folds amplify with greater amounts of shortening, the zone of shear becomes increasingly concentrated beneath synclines (Fig. 9b), ultimately

leading to a narrow shear zone along the thin remnant salt in these locations and thus near the base of salt even beneath the anticlines (Fig. 9c).

    The schematic models of Figures 6, 8, and 9 have relatively little bedding-parallel shortening. As the amount of contractional strain increases, intrasalt deformation is likely to get more complex in several ways. First, compression may lead to increased fluid pressures in siliciclastic or carbonate layers and, consequently, in-situ hydraulic fracturing, as





suggested for the Katanga Copperbelt (Cailteux et al., 2018). Second, strong layers may become disrupted beneath synclines, as is shown on the left side of Figure 7a, with increased amounts of folding and disruption in the anticlines due to lateral salt flow. Thus, deeper levels of salt, which tend to be removed from synclines earlier, will be more transparent in fold cores even if they contain common strong layers (e.g., see Fiduk and Rowan, 2012, fig. 5 or Jackson et al., 2015, figs 6 and 7).

Third, as the salt-cored anticlines tighten over time, the cross-sectional area in the core, which initially increases, begins to decrease until the limbs are roughly parallel (Dahlstrom, 1990; Stewart, 1999). This expels salt, which probably flows along strike into those parts of the anticline that are still in relatively immature, area-increasing phases of shortening, thereby creating a greater likelihood of competent layer disruption in three dimensions. Fourth, thrust faults may develop in the cover with increased amounts or rates of shortening (or due to the salt budget; Stewart, 1999), also leading to increased rupturing

of intrasalt strong layers (e.g., Fig. 7a).

### 3.3 Differential loading

In a sense, the internal deformation generated by differential loading is similar to that typifying contraction: salt moves laterally from areas where it is being thinned to areas of thickening. In differential loading, the synclines and anticlines of shortening are replaced by withdrawal basins and salt pillows. The key difference, however, is that in differential loading,

there is no net shortening of the salt and its cover, which leads to a rather different internal deformation style.

In plane strain, salt deflation beneath overlying depocenters is effectively pure-shear vertical flattening, while the corresponding inflation of salt pillows comprises convergent flow and intrasalt shortening. Thus, intrasalt strong layers form boudins beneath depocenters but are complexly folded within pillows (Fig. 10), with no net change in bed length. If there is little to no mixing of ductile material from above and below each competent layer, shallower layers exhibit more strain

because there is progressively less vertical displacement in the salt with increasing depth (Fig. 10c). With more deformation, of course, the internal deformation becomes more complex than in the simple models shown (see also the model derived from the Oman diapirs in Reuning et al., 2009).

Two-dimensional numerical models with a single intrasalt competent layer illustrate the geometries that develop during differential loading. In a symmetric scenario, salt flows into the pillow from both directions, generating boudinage and

folding or, in effect, convergent intrasalt linked extension and contraction (Fig. 11a). In an asymmetric model of loading only from one direction, with ultimately much greater amounts of net salt movement, the competent layer is almost completely absent beneath the depocenter, and initial folding is followed by complexly disrupted geometries with several stringers on top of each other (Fig. 11b). An example from the northern Netherlands shows that the Z3 anhydrite/carbonate/shale layer of the Zechstein salt (Fig. 1c) forms boudins (stringers) beneath minor depocenters but is

more continuous and folded within the intervening salt pillow (Fig. 11d).

In the case of linear bodies of inflated salt, the two-dimensional analysis of Figure 10 is a reasonable approximation of salt flow for all but the ends of the salt ridge. However, for circular to elliptical pillows (or diapirs), intrasalt deformation is more complex due to convergent flow and consequent constrictional strain (Van Gent et al., 2011). Thus, whereas deeper





flanks are characterized by boudinage on dip lines and folding on strike lines, folds are likely to dominate on lines of any orientation over higher parts of the pillow (Fig. 12).

### 3.4 Passive diapirs

Passive diapirs, which are those that grow at or just beneath the ground surface or seafloor as sediment is deposited, may be

triggered by various means: extension (reactive diapirism of Vendeville and Jackson, 1992), contraction (especially if the roof is thinned by erosion; Coward and Stewart, 1995), or differential loading (e.g., Ge et al., 1997). These early processes, of course, will cause deformation of intrasalt strong layers prior to passive diapirism, whether in the form of boudinage or folding/thrusting. However, in the analysis below, we consider the simplified scenario in which a passive diapir initiates without any precursor deformation.

In such a case, the deformation is similar to that of differential loading in that the salt thins beneath minibasins and thickens in the diapir. However, the strains are much higher: whereas a salt pillow generally has a height to width ratio of less than one, mature passive diapirs are typically at least several times higher than they are wide. Thus, the amount of lengthening of intrasalt strong layers is considerably greater, leading to more boudinage. Contractional folding is less prevalent, except for that produced by constrictional strain in circular to elliptical diapirs as competent beds are caught up in

curtain folds within the ductile halite (see Talbot and Jackson, 1987).

Consider a simple two-dimensional model (Fig. 13a) in which all intrasalt layers experience the same amount of lengthening (although other scenarios are possible, for example when deeper levels are less involved in flow into the growing diapir). Extension (boudinage into stringers) will increase as the diapir grows. The extension $e$ is calculated as the change in length divided by the original length:

$$e = ((l_0 + 2h) - l_0)/l_0 = 2h/l_0,$$

where $h$ is the height of the diapir and $l_0$ is the original length of the strong layers that move into the diapir (Fig. 13a). If we define the aspect ratio of the diapir, $a$, as $h/w$, where $w$ is the diapir width, and the salt fetch factor, $F$, as $(l_0-w)/w$, i.e., the ratio between the width of the area of salt that moves into the diapir and the width of the diapir, then

$$e = 2a/(F+1).$$

In other words, the lengthening of an intrasalt strong layer increases as the diapir grows (as $a$ increases), but decreases as the width of the area of salt that feeds the diapir increases (Fig. 13b). Tall diapirs fed locally experience more internal extension than short diapirs fed from a broader area.

As strong layers are broken into stringers and carried up with the ductile halite within the growing diapir, they rotate to near vertical due to shear caused by upward Poiseiulle flow (Fig. 13c; see also Fig. 2c). In the Kłodawa diapir of Poland, for

example, vertically oriented boudins of anhydrite, dolostone, and clays indicate at least 30% extension (Burliga, 1996), and the largest carbonate/siliciclastic boudins are rotated to vertical near the margins of Onion Creek diapir in the Paradox Basin (Hudson et al., 2017). Because steep stringers are difficult to image on even the best modern seismic data, passive diapirs



typically appear transparent whereas the stringers tend to be more visible where subhorizontal in the deep source layer (Fig. 14). We return to this topic later in the Discussion.

As already stated, more complex strain histories than that shown in Fig. 13a are possible. Instead of the entire salt layer flowing synchronously into the growing diapir, shallower levels may move upward first, followed only later by salt from 5    deeper levels. This has the effect of leaving younger stringers near the margin of the diapir at deeper levels, whereas older stringers will be concentrated in the center of the diapir and at shallower levels. These patterns are observed in exposed diapirs, for example in Iran (e.g., Jackson et al., 1990) and South Australia (T. Hearon, pers. comm.)

The development of allochthonous salt (other than source-fed thrusts, Hudec and Jackson, 2006) is a form of passive diapirism in which the salt moves more laterally than vertically at the ground surface or just beneath the seafloor. Moreover, 10    most salt sheets originate from steeper feeder diapirs in which the strong layers are already disrupted. With more lateral flow, stringers rotate to lower angles than in the steep diapirs during flow subparallel to the base salt. Although it might be thought that stronger, denser intrasalt layers are left behind as the more ductile halite and bittern salts flow laterally in the salt sheet, that is demonstrably not the case. Salt sheets in South Australia contain abundant stringers of carbonates, siliciclastics, and igneous rocks that can be more than 2 km long and hundreds of meters in thickness (see maps in Rowan et 15    al., 2019).

## 4 Discussion

We have presented an overview of the deformation of competent intrasalt layers in different modes of salt tectonics. For the sake of simplicity, the focus has been on individual competent beds, but of course there can be larger packages comprising multiple competent layers or even non-evaporite or evaporite-poor strata within the overall LES. Moreover, we have 20    emphasized the deformation of individual beds well within the LES; those closer to the margins (upper or lower) do not deform as readily even in strong salt flow. In the following sections, we discuss the definition of the top and base LES, the roles of layer thickness and density, the mechanical stratigraphy of multilayers, and spatial or temporal variations in the operative mode of deformation.

### 4.1 Definition of LES

25    The mobile portion of an evaporite sequence does not necessarily represent the entire LES. In many cases, the base of the evaporites is dominated by interbedded anhydrite and carbonate or siliciclastics, with no or little halite or bittern salts (e.g., the Santos and Oman salt layers, Fig. 1a, b), due to the gradual and cyclical onset of evaporative conditions. These stronger margins may or may not be involved in the deformation. If they are, the deformation may be spatially variable, so that on seismic data, the base of transparent salt appears to cut up and down section (Fig. 15). Where the base is shallower, it is 30    underlain by an in-place package of bright reflectors (dominated by anhydrite with carbonates/siliciclastics, depending on the salt basin); where the base of the mobile salt is deeper, the bright package is absent. Stringers near the base of salt are



interpreted to represent the same basal package where it has been incorporated into the mobile salt. In addition, the transparent areas likely contain stringers that are too small or steep to be imaged.

One consequence of stronger basal layers in the LES is that even after the mobile portion is removed, for example by flow from beneath a minibasin into a flanking diapir, evaporites may remain along the resultant weld. In the Santos Basin, a

well encountered 22 m of anhydrite, carbonate, and sandstone along a seismically defined weld (Jackson et al., 2014), and 31 out of 41 wells that penetrated thin salt (<100 m thick) in the Campos Basin encountered only anhydrite (Wagner and Jackson, 2011). A common interpretation is that these represent broken-up blocks of less mobile material trapped along the weld due to boundary drag effects as the salt thinned. We suggest as an alternative that some of these represent the basal portions of the LES that were never broken up and thus remain stratigraphically in place.

The tops of some LES may also have a relatively thin anhydrite- and/or carbonate-dominated section (Fig. 1a, b) that may or may not be involved in the deformation. In other basins, there can be thicker sequences of siliciclastics and/or carbonates interbedded with variable amounts of evaporites overlying a more halite-rich salt layer. In the Red Sea salt basin, for example, the Mansiyah/South Gharib salt is overlain by the Ghawwas/Zeit formations. In the proximal Midyan area of the NE Red Sea, the Ghawwas Formation comprises mostly siliciclastics with thin anhydrites (Hughes and Johnson, 2005)

and clearly forms the brittle overburden to the ductile Mansiyah salt (see seismic profiles in Tubbs et al., 2014). Farther basinward, however, the relative proportion of siliciclastics in the Ghawwas Formation decreases and there are increasing amounts of interbedded halite (K. Śliż, pers. comm., 2018). Here, the Ghawwas Formation deforms with the Mansiya salt as part of the mobile LES, with variable amounts of deformation including recumbent, isoclinal folds similar to those observed in the Santos Basin (Rowan, 2014, figure 9a). Thus, the top of the mobile LES shifts stratigraphically up-section from

proximal to distal regions.

Another example is provided by the Precaspian Basin. The main Kungurian salt (Fig. 1d) is overlain by Ufimian and Kazanian interbedded mudstone and halite, with the combined package considered by some as the LES (Gralla and Marsky, 2000; Volozh et al., 2003). However, there are variable relationships: whereas both the Kungurian and Kazanian strata together form diapirs in more distal areas (Volozh et al., 2003, figure 5, panels B and E), the Kazanian sequence forms the

brittle overburden to the ductile Kungurian salt in proximal areas dominated by extension (Volozh et al., 2003, figure 5, panels A, D1, and D2). Recent interpretations based on 3-D seismic data interpret the top of Kungurian strata to be the top salt, with the younger successions representing minibasins that become encased in salt due to subsidence and subsequent flow of mobile salt over their tops (Fernandez et al., 2017; Duffy et al., 2017). Interestingly, the location of these possible encased minibasins is between the proximal and more distal domains of Volozh et al. (2003). We suggest that there is a

complete spectrum of relationships that is dependent largely on the relative proportions of evaporites and more competent layers within the Uffimian and Kazanian sequences. Where halite is absent or rare in the younger section, these strata form brittle minibasins above the Kungurian salt, but where halite is common in the younger section, all three sequences behave as a ductile LES. In between, thicker packages of relatively halite-poor strata still form part of the LES, but in a more coherent manner that appears analogous to suprasalt minibasins that become encased.





A more speculative example is provided by the combination of the Permian and Triassic salts of the Southern Permian Basin of Europe. The Zechstein Group is well established as the primary mobile salt interval, but there was also halite deposition within the Buntsandstein, Muschelkalk, and Keuper sequences of the Triassic, with the main intrasalt sequences being the Buntsandstein siliciclastics and the Muschelkalk carbonates (e.g., Beutler, 1998; Geluk et al., 2000; Geluk, 2007b;

McKie, 2017). The aerial and thickness distribution of Triassic halites was highly variable, which most likely influenced the local deformation. In NW Germany, for example, blocks of the lower and middle Buntsandstein separated by gaps have been interpreted as rafts formed during thin-skinned extension decoupled from presalt extension in other locales (Mohr et al., 2005). However, it is possible that some of the deformation represents intra-LES boudinage during not just extension but also salt evacuation and diapirism. If this were the case, there might be age-equivalent stringers within the taller diapirs that

are unrecognized due to greater amounts of disruption and steeper attitudes.

In the Santos Basin, the onset of salt flow has been interpreted to have occurred during ongoing evaporite deposition (e.g., Davison et al., 2012; Quirk et al., 2012). However, detailed stratigraphic analyses demonstrate that the LES is mostly prekinematic (Fiduk and Rowan, 2012; Jackson et al., 2014, 2015): the internal stratigraphic intervals remained parallel and undeformed, with only regional thickness changes, until near the cessation of evaporite deposition. In other basins, however,

the upper portions of the LES were indeed deposited while deeper, older portions were already moving due to extension or contraction. This is interpreted, for example, in the Zechstein salt basin (e.g., Stewart and Clark, 1999; Geluk, 2005; van Gent et al., 2011) and also in the Miocene salt of the northern Carpathians (Kolasa and Ślączka, 1985; Bukowski, 1997). Other areas where synkinematic evaporite deposition occurred include the Red Sea and Precaspian basins.

One implication of the different timing relationships concerns the interpretation of stratal thinning within the LES.

Thickness variations may represent growth strata deposited during movement of underlying salt, as is typical for suprasalt minibasins. In contrast, they may be purely structural, caused by differential flow within the salt. And of course, there can be any combination of depositional and structural thinning of intrasalt strata. It is not easy to discern the difference: in the Levant Basin, apparent growth geometries are interpreted as evidence for shortening during evaporite deposition (Gvirtzman et al., 2013) but also as being due entirely to structural thinning during post-Messinian shortening (Allen et al., 2016).

**4.2 Competent layer properties**

We have focused on the deformation of intrasalt competent beds, treating them simply as anomalous layers within the power-law-fluid halite. These strong layers are not all equal – they can differ in strength, thickness, and density. Below, we discuss the effects of varying these properties. We also address the mechanical stratigraphy of multilayers, i.e., the relative proportions and thicknesses of brittle and ductile layers within discrete stratigraphic packages.



### 4.2.1 Thickness and strength

The wavelength, or spacing, of structures is dependent on layer thickness. Boudins of thicker layers can thus be longer than those of thinner layers. In shortening of strong layers embedded in a weak matrix, the initial wavelength of the resultant buckle folds increases as the layer thickness increases (Ramberg, 1960; Biot, 1961).

As mentioned briefly above, the competency contrast between relatively strong layers and weak halite and bittern salts influences the style of deformation. In extension, if the competency contrast between the strong layer and its more ductile matrix is relatively small, pinch-and-swell structures (or drawn boudins) develop; if it is relatively large, the strong layer extends by brittle fracturing and forms torn boudins (e.g., Ramberg, 1955; Ramsay, 1967; Goscombe et al., 2004; Abe and Urai, 2012). The large contrasts within LES result in predominantly torn boudins (Figs. 2b and d, 5). Interestingly, thinner

layers tend to rupture less during layer-parallel extension than thicker layers (von Hagke et al., 2018).

Differing competency contrasts also impact the mechanics of buckle folds and the consequent structural styles in shortening (e.g., Ramsay, 1982; Ramsay and Huber, 1987). When the competency contrast is low, the initial wavelength is short relative to layer thickness, the layer thickens due to layer-parallel shortening (LPS), and low-amplitude cuspate-lobate folds result. When the contrast is high, the initial wavelength is long relative to layer thickness, LPS is negligible, and high-

amplitude buckle folds and ultimately ptygmatic folds develop. Again, the typical large competency contrasts within LES mean that buckle and ptygmatic folds are predominant (Fig. 7a and b), but relatively weak non-evaporite layers such as overpressured mudstones are likely to provide exceptions.

### 4.2.2 Mechanical stratigraphy of multilayers

The discussion of the role of layer thickness and strength above was for single competent beds encased in a weak matrix.

Typically, however, an LES can be considered as a multilayer, i.e., an interbedded sequence of rocks with different thicknesses, rheologies, and competency contrasts. Here, we focus on buckling because of the rich literature on folded multilayers (e.g., Ramberg, 1961, 1964; Ramsay and Huber, 1987; Hudleston and Treagus, 2010); presumably, there are analogous but different effects in extended multilayers.

There are numerous controls on fold geometry, the most important of which for intrasalt deformation are: 1) the rheology

of each layer and the contrast in rheology between layers; 2) the thickness of each layer and the grouping of layers into packages of relatively low or high competence; 3) the nature of the interfaces between layers, i.e., whether layers are effectively welded together (which is typically not the case for salt); and 4) the geometry of the bounding presalt and suprasalt strata. These combine to determine different styles of folding based largely on the relative proportions and thicknesses of strong and weak layers. If the strong layers are far enough apart that there is no interference of the ductile

material adjacent to one and that adjacent to another, each folds independently and disharmonic folds result (Figs. 6b, 7b and d; Fiduk and Rowan, fig. 9). When the weak layers are relatively thin and of similar thickness, the competent-layer thicknesses also don't vary too much, and the competence contrast is fairly consistent, the folding is largely harmonic (Fig.




7e; Fiduk and Rowan, fig. 6; Jackson et al., 2015, figs. 7a and 10a). However, if the ductile layers are relatively thin and the thicknesses and competency contrasts of the strong layers vary markedly, polyharmonic folds develop in which folds or thrusts of thinner competent layers are themselves folded by thicker, even stronger layers. This can also occur due to the longer-wavelength structures of the overburden deforming shorter-wavelength folds within the LES (Fig. 17, basinward end).

The mechanical stratigraphy of the LES multilayer also influences how internal strain is partitioned. There may be multiple detachment levels (Cartwright and Jackson, 2008) by localized shear in intervals of thicker halite and/or bittern salts. The thicker weak levels also experience more Poiseuille flow than the intervening stronger packages (Cartwright et al., 2012).

### 4.2.3 Density

The effects of the strength and thickness of competent layers embedded in a ductile medium, as discussed above, are well established and accepted. However, the effects of density contrasts are less well understood and thus somewhat more controversial. Despite most intrasalt strong layers being more dense than the encasing halite (e.g., anhydrite, carbonates, most siliciclastics, volcanics), they have been carried up in diapirs and into allochthonous sheets by the mobile halite.

Examples are found in salt basins throughout the world, including the Flinders Ranges of South Australia, the Oman salt basins, the Zagros Mountains of Iran, the Spanish Pyrenees, and La Popa Basin of Mexico. Analog and numerical models have suggested that once the salt movement slows or ceases, the more dense stringers sink back down through the halite (Koyi, 2001; Chemia and Koyi, 2008; Chemia et al., 2008). However, this is questioned by Van Gent et al. (2011), based on a lack of observed correlation between stringer size and depth, and numerical simulation by Li et al. (2012b) helps show why

this doesn't occur. The explanation lies in the complex cyclic rheology of halite deformation. As mentioned above, differential stress associated with diapiric flow is circa 2 MPa, with roughly equal contributions of pressure-solution creep and dislocation creep (Fig. 3a); that caused by density differences is about 0.1 MPa. Thus, diapiric flow can carry more dense stringers upward and sinking is insignificant. Once flow stops, the differential stress relaxes and pressure-solution creep becomes dominant. However, the grain boundaries 'freeze' and the fluid films form bubbles (Urai et al., 1986), so that

pressure solution is switched off and sinking ceases. This is the case at least for the relatively small Zechstein stringers, but these stresses can rise to several MPa for large dense intrasalt packages, such as those in the Precaspian Basin.

Do density contrasts play any role in determining structural styles during ongoing deformation? Folding of strong layers driven by progradational loading is shown to be influenced by varying densities: synclines sink more in the case of more-dense layers, whereas anticlines rise higher for less-dense layers (Albertz and Ings, 2012). In contrast, other numerical

models produce no such sinking of more-dense anhydrite analogs (Li et al., 2012a; Raith, 2017). Dooley et al. (2015) built an analog model of a deeper pure polymer (halite) overlain by interbedded polymer and stronger and more dense granular interlayers (anhydrite). When a differential load was applied, the deep polymer level (their A1) formed internal 'diapirs' and 'salt sheets' while the more dense upper portion (their A2-A4) sank and formed recumbent, isoclinal folds (Fig. 16a). They





ascribe this geometry to intrasalt Raleigh-Taylor overturn of two viscous materials with a density inversion combined with folding generated by allochthonous salt emplacement. This model is used to explain the observed deformation in some complex structures of the Santos Basin (Fig. 16b), interpreted to have grown by passive diapirism followed by minor late-stage shortening (Jackson et al., 2015). However, it should be kept in mind that the model material used was Newtonian

viscous, in which sinking driven by density differences is much faster than in a material that deforms by power-law creep and has a much higher effective viscosity at low differential stress, i.e., one correctly scaled for salt tectonics.

In contrast, Fiduk and Rowan (2012) use a depth-migrated version of the same data from the Santos Basin to suggest that the structures are primarily contractional. Here, we suggest that the analog model of Raleigh-Taylor overturn (Fig. 16a) cannot explain the observed structures (Fig. 16b). In a simple line-length restoration of the model, all layers fit back together

almost perfectly (Fig. 16c), as expected given the initial configuration and the applied boundary conditions (vertical differential loading). However, a line-length restoration of the real example shows several problems (Fig. 16d). First, the deepest layer restores to a length 3 km longer than the deformed-state section. One possible explanation is that the beds were lengthened during folding, but there is usually little observed correlative thinning in this and similar structures. Another possibility is that the approximate depth geometry is significantly off, but the generally excellent images of the presalt

section indicate that this is unlikely. We suggest instead that there was at least 3 km of intrasalt shortening within this single structure. There are folds and thrusts of the overburden that extend away from the structure (see Jackson et al., 2015, figure 9b), but without as much shortening; as explained above, a higher level of shortening within the salt is to be expected. Second, the shallower beds all have shorter restored lengths and, moreover, the lengths are unequal. Yet both Fiduk and Rowan (2012) and Jackson et al. (2015) agree that the initial configuration comprised parallel strata of equal lengths. The

amount of missing section ranges between 1.9 and 5.7 km, with no systematic pattern up-section. In short, the data do not match the model.

The observed geometries in the Santos Basin are also incompatible with numerical models and other real examples of inflation and passive diapirism caused by differential loading, in which internal strong layers form boudins due to the required lengthening and high viscosity contrasts (Figs. 10 to 14). Therefore, we offer an alternative interpretation that has

several key components. First, the structure of Figure 16b was formed primarily by shortening, with only a minor component of passive diapirism due to erosional and extensional unroofing of the top of the structure. Second, the overall internal geometry is that of a breached ptygmatic fold, albeit with some disharmonic deformation. Third, the deep core of the structure is indeed made up of the deepest layer of the LES, but represents the salt core of a contractional anticline rather than a feeder diapir to a shallow sheet. Fourth, the upper, mostly transparent part of the structure is not an allochthonous

sheet made up of the A1 layer, as in the analog model; instead, it represents the upper portion of the anticline, which contains a mix of A1 and the missing portions of the shallower layers, but with bedding disrupted so that there is no seismic coherency. Finally, there is no hard boundary between layered and transparent seismic facies, but rather a gradual transition that we interpret as a progression from well-behaved bedding to slightly disrupted strata to a melange of strong pieces in a matrix of ductile material.





Interestingly, the synclinal portions of the structures tend to be better imaged than the anticlinal portions, implying that rupturing of the strong layers is more pronounced within the anticlines. This is incompatible with established models of folding, in which the neutral surface (boundary between local tensile and compressional stresses caused by bending of strong layers) migrates to the insides of folds, i.e., the bases of anticlines and the tops of synclines (e.g., Frehner, 2011). This puts

stronger layers within both anticlines and synclines in local tension, which should lead to equal extensional breakup of bedding. We suggest that the explanation may lie in the mechanical stratigraphy of the multilayer. If the upper part of the folded stratigraphy contains more weak material (halite and especially bittern salt) and the lower part has higher-viscosity layers (as is the case for layers A2-A4 of Fig. 15b; see well logs in Jackson et al., 2014, 2015), thin strong layers within the upper portion are more likely to be stretched and broken into boudins and thus more poorly imaged. To sum up, although

density contrasts may have contributed to the deformation, for example by aiding in the breaching of the early anticline, we interpret the role as minor.

### 4.3 Combined modes of salt tectonics

We have shown that the internal deformation of salt sequences with stronger interbeds varies depending on the mode of salt tectonics. Specifically, we focused on extension, contraction, differential loading, and passive diapirism. However, areas that

experienced exclusively one mode are rare. In salt movement triggered by extension or contraction, for example, differential loading plays a contributing role because some areas of salt rise while others sink and develop greater thicknesses of overburden. Passive diapirs usually have an early phase where salt movement was initiated by extension, contraction, or differential loading. Moreover, it is common for passive diapirs, whatever their origin, to experience late extension or contraction since they are the weakest portions of basins and thus tend to localize strain.

Nevertheless, we suggest that using the models presented here can aid in distinguishing between and understanding different styles and processes of salt-related deformation. In the following sections, we use two salt basins to illustrate and examine more complex scenarios. First, we use the Santos Basin of the Brazilian passive margin to show how the dominant style varies spatially, from proximal extension and passive diapirism to distal contraction. Second, we discuss the temporal evolution of salt tectonics in the Southern Permian Basin of Europe, from early extension to differential loading and passive

diapirism and to late contractional rejuvenation.

### 4.3.1 Santos Basin

The northern half of the Santos Basin is dominated by two major salt-tectonic provinces, the mostly evacuated and welded salt of the proximal Albian Gap and the inflated and folded salt of the distal São Paulo Plateau (e.g., Modica and Brush, 2004; Quirk et al., 2012; Garcia et al., 2012; Kukla et al., 2018). We use a modern depth-migrated 3-D seismic profile to,

first, describe the large-scale structural styles and domains and, second, describe the associated intrasalt deformation with reference to the models presented in this paper.





Starting at the proximal, NW end of the line and moving basinward, we identify four domains (Fig. 17). First, landward of the a major step in the base salt over the Merluza fault is an expulsion-rollover structure and a vertical, symmetric passive diapir. This is an area of differential loading and diapirism, with little lateral translation because of the barrier to basinward movement provided by the 2.6 km offset in the base salt at the Merluza fault. Second, on the basinward side of the step is a

domain dominated by extensional diapirs and faults with associated extensional rollovers (see also Fig. 5c). Faults and equivalent diapir margins dip both basinward and landward, including the large counterregional extensional system of the Albian Gap. Third, basinward of the Albian Gap is an area of passive diapirs and synclinal minibasins, but with low-angle stratal truncations and minor components of shortening, especially for the more distal diapirs. We interpret this as a transitional zone between proximal extension and distal contraction. Finally, the basinward end of the line comprises a series

of anticlines and synclines over salt that is 0.4 to 1.2 km thick, and is the updip portion of the major São Paulo contractional domain that extends farther basinward (see also Fig. 7e).

A quick glance at the seismic profile shows that the internal character of the salt varies systematically depending on the mode of salt tectonics (Fig. 17). This is despite the fact that the entire line is in the same local part of the salt basin, where the evaporites were originally layered throughout the area. The tall passive diapir just landward of the Merluza fault is

transparent, although there are several stringers visible on the lower portion of the proximal flank. Note also the subtle extensional geometries within the almost welded salt landward of the diapir, presumably related to thinning and stretching of the salt during flow into the diapir. Moving basinward, the salt rollers and reactive diapirs of the extensional domain are also largely transparent due to disruption of the intrasalt layers by boudinage (Figs. 4 and 5). The diapirs of the transitional domain, in contrast, are characterized by more internal reflectivity, with discrete packages of complexly folded strata but

also significant disruption. This is not typical of passive diapirs, but then these are not tall, narrow stocks or walls – they are generally broad salt bodies with low-angle stratal truncations on their flanks. One possibility is that the contractional component of the deformation resulted in internal folding and thus greater preservation of layer coherency, rather than the lengthening and boudinage characteristic of simple passive diapirs. Another is that these are effectively salt pillows where the salt was emergent rather than covered by the oldest suprasalt strata, and salt pillows are characterized by disrupted folds

(Figs. 10 to 12). Finally, the distal contractional domain is marked by largely coherent and folded intrasalt layers, but with thickness variations possibly due to differential flow of more ductile portions of the salt from beneath synclines into anticlinal cores. The overall pattern in this area is typical of salt sequences that have experienced minor to moderate amounts of shortening (Figs. 6 and 7). Note that the more complex structure of Figure 16b is located slightly more basinward within this contractional domain. Again, salt bodies dominated by contraction tend to have more coherent intrasalt strata, whereas

passive diapirs driven by differential loading have disrupted, rotated strong layers and thus more transparent seismic character.





### 4.3.2 Southern Permian Basin

Just as the same salt basin can exhibit spatial variations in the dominant mode of salt tectonics, individual salt structures commonly experience temporal variations in the dominant mode. An excellent example is the Southern Permian Basin of northern Europe, where the Permian Zechstein salt was affected by a complex and multiphase tectonic history. Rifting began

even during evaporite deposition (e.g., Geluk, 1999; Geluk et al., 2007; Biehl et al., 2014) and continued at different stages during the Triassic (e.g., Mohr et al., 2005; Geluk, 2007a; Rowan and Krzywiec, 2014). Because extension initiated local subsidence, consequent differential loading became a contributing driver of salt movement, with a combination of extension and loading generating a series of salt pillows, diapirs, and minibasins. The Alpine Orogeny was then responsible for contractional modification and rejuvenation of the salt structures during the Late Cretaceous and into the Cenozoic (e.g.,

Baldschuhn et al., 1991; de Jaeger, 2003; Krzywiec, 2006).

A 3-D depth-migrated seismic profile from the Groningen High of the Netherlands (Raith et al., 2016) illustrates some of the resulting geometries. It crosses two pillows: the Slochteren Pillow is in the common footwall to west-dipping and south-dipping presalt extensional faults, and the Veendam Pillow is located partly over the southern flank of an E-W trending graben and partly over the graben itself (Fig. 18). Although folding of all strata over both pillows records late contractional

growth, their early histories were different. The Buntsandstein (oldest postsalt sequence) is highly asymmetric on the Slochteren Pillow, with thinner strata to the north (to some extent erosional beneath the base-Cretaceous unconformity) and thicker strata to the south suggesting early, partially decoupled extension in the footwall of the underlying south-dipping presalt fault. In contrast, the Buntsandstein on the Veendam Pillow is largely symmetric, with thinning showing that there was syndepositional inflation. This was likely due to differential loading, triggered by formation of the graben between the

two presalt faults and consequent subsidence of the suprasalt depocenter between the two pillows (see modeling by Warsitzka et al., 2015).

The resulting intrasalt deformation is correspondingly broadly similar but also somewhat different within the two pillows (Fig. 18a). The Z3 intrasalt layer geometry mimics the top-salt geometry, which resulted from the combined effects of extension, subsidence and inflation driven by differential loading, and late contraction. However, it is broken into stringers,

with intrasalt extension somewhat more pronounced within the extensional Slochteren Pillow and beneath the intra-pillow depocenter. The Z3 layer is more continuous within the Veendam Pillow, which formed due to early inflation and late contraction.

### 4.4 Implications

The nature of the deformation of intrasalt strong layers, and thus their distribution within the salt, has implications for

several aspects of imaging and drilling through layered evaporite sequences as well as for interpreting and modeling salt-related deformation. Below, we touch briefly on such issues as seismic and well interpretation, building velocity models, and drilling risks.



### 4.4.1 Well data

Wells drilled into and through LES can show apparently significant variations in the percentages of different lithologies encountered. For example, in a small area of the Santos Basin, Brazil, six wells show large variations in the proportions of halite, anhydrite, and bittern salts (Jackson et al., 2014, 2015). This corresponds to differences in the average densities of

specific intrasalt layers, which was used to explain contrasting structural styles. A larger study of 26 wells in the same basin found similar variations in evaporite proportions (Gonzalez et al., 2016).

Local differences in original LES composition are to be expected when older parts of the salt are moving as younger portions are deposited, especially when siliciclastic rocks are interbedded with the evaporites. They are also expected on a regional scale because anhydrite is more prevalent and thicker in marginal areas of salt basins and halite and bittern salts are

more common in basin centers (Warren, 2016). Significant local variations, however, are not expected in this portion of the Santos Basin, where there is general agreement most or all of the LES is prekinematic (Gamboa et al., 2008; Fiduk and Rowan, 2012; Jackson et al., 2014, 2015). Moreover, evaporite deposition at any one time cannot change from one mineral to another laterally since vertical boundaries between different brine chemistries are impossible (Warren, 2016). Thus, local variations in proportion seen in wells are most likely due to deformation. Even where stronger layers within the LES largely

maintain their coherency and are therefore well imaged on seismic data, preferential flow of more ductile salts will lead to structural thinning and thickening.

In short, one-dimensional samples through deformed evaporites should not be used to determine original proportions, densities, and stratigraphy. This is especially true in the case of diapirs, where, as explained above, competent beds get folded and broken into boudins and rotated to vertical. Thus, there is a sampling bias encountered in vertical wells: a stringer

100 m long and 10 m thick is more likely to be penetrated where it has low dip in the original salt layer than where it is near vertical in the diapir. Wells drilled into steep diapirs will therefore encounter higher proportions of halite than those drilled in other structures, except of course if a vertical well happens to drill along a vertical competent layer.

### 4.4.2 Seismic data

The same issues impact seismic imaging within salt structures, as discussed, for example, in the Santos Basin example (Fig.

17). Where the strong layers are disrupted, the image is degraded due to reduced seismic coherency. Again, this is especially true within steep diapirs because of more intense deformation and the near-vertical orientation of most stringers (see Jones and Davison, 2014). Thus, diapirs typically appear transparent even when other parts of the salt system display internal seismic reflectors. Seismic transparency does not imply that the diapirs are mostly or exclusively composed of halite; for example, tall diapirs in Oman have little to no internal reflectivity, yet surface exposures show that stringers are common

(compare figs. 3b and 5 in Reuning et al., 2009). In any given area, diapirs, pillows, salt rollers, the cores of contractional anticlines, etc. are generally not fundamentally different in their composition, just in their seismic expression. However, this





is true only for salt layers that are prekinematic; if LES deposition continues after the onset of salt-related deformation, there can be significant lateral variations in composition and thus seismic character.

In any case, the variability in seismic expression due to differences in the dominant mode of salt tectonics has several ramifications. First, it impacts the building of velocity models used in depth migration and imaging of sub- or presalt
geometries. Two end-member options are to use a single salt velocity everywhere or to use one velocity for transparent areas, which are presumed to be halite, and another velocity for layered seismic facies. Neither of these is likely to be correct. Even approaches that combine seismic images with lithology and velocity data callibrated from wells (e.g., Barros et al., 2017) are liable to be inaccurate because the transparent areas typically include stringers of non-halite lithologies that are not imaged. In other words, the transparent and layered areas usually have less and more halite, respectively, than thought.

Second, the seismic image can be misleading to those planning and drilling wells through the salt. It is usually safer to drill through areas where there are no competent layers (see below), and wells are planned accordingly. However, the absence of an imaged stringer does not necessarily mean that the stringer is truly absent. In the Southern Permian Basin, some wells have penetrated the Z3 stringer even when it is not visible due to small size or steep dips (Van Gent et al., 2011).

Third, seismic data can be misinterpreted, with sometimes costly consequences. For example, layered evaporites may be
misidentified as non-evaporite sequences and may be drilled if in a trap geometry. We know of one case in which a large turtle structure between two obvious diapirs was tested. It turned out to comprise about 70-80% evaporite and 20-30% mudstone, with no reservoirs and no hydrocarbons. Alternatively, transparent diapirs may be misinterpreted as consisting purely of halite. In one instance, gravity models run using a standard halite density predicted that a salt diapir in the Nordkapp Basin had large overhangs and a narrow feeder, providing room for a subsalt trap. However, a well and its
sidetrack both remained in salt, encountering caprock and stringers of anhydrite, carbonate, sandstone, and mudstone that were not imaged. Post-drill modeling using a higher density that reflected the contributions of anhydrite and carbonate supported the diapir being wider than originally thought (Stadtler et al., 2014).

### 4.4.3 Drilling hazards

Stringers can sometimes be exploration targets, such as the interbedded carbonate reservoir and source rock layers within the
Ara salt of Oman (e.g., Peters et al., 2003; Reuning et al., 2009). However, the majority of wells in basins with sub- or presalt plays try to avoid stringers while drilling through the salt. Both fluid kicks and losses have been encountered, for example, while drilling through stringers in the Zechstein salt (Strozyk, 2017). Highly fractured carbonates can be overpressured due to the surrounding sealing salt, but normal pressures may be encountered if the fluids have had a chance to escape, for example by contact with the suprasalt strata (e.g., Kukla et al., 2011). Again, though, stringers are not
necessarily imaged, especially in more deformed salt, presenting an unavoidable drilling risk.





## 5 Conclusions

Layered evaporite sequences comprise different evaporites and, typically, non-evaporite layers. Most studies on intrasalt deformation have focused on halite as the dominant evaporite mineral. Here, we address instead the deformation of relatively strong layers such as interbedded anhydrites, carbonates, and siliciclastics. The effective viscosity contrasts between these
strata and ductile halite and bittern salts range from 1 to over 5 orders of magnitude. As a result, the deformation of the strong layers is considerably different than the ductile flow of the more mobile evaporites. In shortening, competent intervals tend to form concentric folds formed by bedding-parallel shear along the weak layers. In extension, they behave as brittle materials, breaking into stringers due to boudinage.

The layered rheological contrasts influence how salt behaves in different modes of salt tectonics. In extensional settings,
boudinage of strong layers dominates. Stringers are subhorizontal in weakly extended areas but become more complexly deformed as extension increases, such as in salt rollers in the footwalls of normal faults. Deeper strong layers may be more extended than shallow ones. In contrast, partly because rocks are stronger in contraction than in extension, competent layers tend to maintain coherency in low to moderate amounts of shortening. They form variable fold geometries (more rarely thrusts) ranging from symmetric anticline-syncline pairs to asymmetric ptygmatic folds, with disharmonic and polyharmonic
styles common. Some layers within the LES may experience different amounts of shortening depending on how strain is partitioned. As salt beneath the suprasalt synclines thins and moves into the anticlines, the latter become areas of more intense folding, thrusting, and disruption.

Differential loading of salt generates areas of thinning salt and adjacent areas of salt inflation, with bulk flow from the former into the latter. As a result, boudinage is dominant beneath suprasalt depocenters and folding dominates in salt
pillows. Three-dimensional convergent flow causes an additional component of tangential contraction and folding. If the salt breaks through to initiate passive diapirism, significant lengthening of intrasalt strong layers as the diapir grows in height leads to increased boudinage along with folding, with fold axes rotated to near-vertical due to strong vertical stretching and localized shear in curtain folds. The stringers rotate back to low angles if the salt subsequently advances more laterally in allochthonous sheets.

In some cases, salt may be dominated by a single style of deformation and thus fit one of the end-member models nicely. In many cases, however, more complex histories result in a progression from one mode to another, as in the case of a passive diapir that forms from an early contractional anticline, and the resultant deformation is a hybrid of the simple models. Even when one mode dominates, however, greater magnitudes of deformation generally lead to increasingly complex internal geometries. Moreover, different areas of a salt basin may be characterized by different modes of salt tectonics. This is
especially true of passive margins, where proximal, translational, and distal provinces are dominated respectively by extension, differential loading, and contraction, with resultant diapirs triggered by different mechanisms.

The variable styles and geometries of intrasalt deformation have important implications for interpreting seismic images of salt because they cause varying amounts of disruption of intrasalt strong layers and thus seismic coherency. Competent





strata tend to maintain structural coherency in low to moderate amounts of contraction and are thus more likely to be visible, although steep fold limbs might not be imaged. The same is true, to a somewhat lesser degree, for salt pillows generated by differential loading. Extension, on the other hand, ruptures the layers into boudins: if the amount of extension is relatively minor, the stringers are typically imaged, but the salt is more transparent in salt rollers and drape folds over major presalt

faults. The most transparent areas, however, are steep diapirs due to the large amounts of intrasalt extension, the resultant rupturing of competent beds, and the rotation of stringers to near-vertical.

Similarly, the variable intrasalt deformation has implications for wells drilled through salt. Where intrasalt layers are more coherent, i.e., in contractional salt structures and salt pillows, wells are more likely to encounter the entire LES stratigraphy. However, the thicknesses encountered will rarely represent the original thicknesses owing to differential flow

of more ductile evaporites. Where strata are disrupted into stringers, wells drilled through apparent gaps will typically not encounter that part of the stratigraphy, although small, poorly imaged stringers might still be penetrated by accident. It is in steep diapirs where well results will be the most misrepresentative of the original stratigraphy: relatively thin stringers rotated to vertical will rarely be penetrated, so that the diapirs may be misinterpreted to comprise solely or mostly halite. In all cases, wells are one-dimensional samples through deformed salt and should not be used as indicators of the original

lithology, thickness, and density distribution of the LES.

In summary, deformation of intrasalt strong layers varies considerably, just as the larger-scale salt-tectonic processes differ within and between different salt basins. We have presented an analysis of this variability based on the rheology of different layers within salt sequences, numerical and analog models, and seismic and well data. It is our hope that the ideas offered here will help both academic and industry scientists understand better the fundamental concepts as well as the

practical aspects of both drilling through and seismic imaging of salt structures.

**Author contributions**

The ideas behind this paper were originally conceived by authors MR and CF, arising out of early investigations of the Santos Basin. MR did the bulk of the writing and figure preparation, with JU primarily responsible for the section on rheology and CF and PK contributing throughout the manuscript. All authors declare that they have no conflicts of interest.

**Competing interests**

The authors declare that they have no conflicts of interest.





**Acknowledgements**

First, we thank PGS, through H. Lebit and A. Lewis, for permission to show some of their recent data from the Santos Basin. We are also grateful to T. Dooley, Y. Feng, M. Hudec, C. Jackson, F. Peel, O. Pla, A. Raith, and F. Strozyk for original versions of their figures that were redrafted here. XX and YY provided helpful reviews that improved the paper, and
Midland Valley Exploration is thanked for the MOVE™ software. MR was partially funded by the Salt-Sediment Interactive Research Consortium at The University of Texas at El Paso, run by K. Giles and sponsored by BHP-Billiton, BP, Chevron, ConocoPhillips, ExxonMobil, Hess, PGS, Repsol, and Total. JU and PK acknowledge the German Science Foundation for funding several salt-related projects, NEDMAG for supporting the PhD work of Alexander Raith, and NAM for access to the 3D seismic data from the Netherlands.

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

**Figure Captions**

**Figure 1.** Representative stratigraphic columns through layered evaporite sequences (LES): (a) upper Aptian Ariri Fm. from the offshore
Santos Basin, Brazil (adapted from Gamboa et al., 2008); (b) uppermost Precambrian to Lower Cambrian Ara Gp. from the onshore Oman salt basins (adapted from Peters et al., 2003); (c) Upper Permian Zechstein Gp. from the Dutch sector of the Southern Permian Basin (adapted from Van Adrichem Boogaert and Kouwe, 1994); (d) Lower to Middle Permian Kungurian salt from the central portion of the Precaspian Basin, Kazakhstan (adapted from Gralla and Marsky, 2000); (e) uppermost Miocene Messinian evaporites from the Levant Basin, eastern Mediterranean (adapted from Feng et al., 2016, and V. Robertson, pers. comm.) Although lateral variations in stratigraphy
exist in every example, they are not shown here due to the difficulty of chronostratigraphic correlations in evaporite sequences (see Warren, 2016). Sections (a) and (e) are from wells, whereas the others are more schematic; thicknesses and age spans are not equivalent in the different examples. Labels on the sides of (b), (c), and (e) are intrasalt stratal packages defined in the literature and referred to in the text and figures.

**Figure 2.** Examples of intrasalt deformation. (a) HyMap image with oblique cross-sectional view of exposed Witchelina diapir in the
Willouran Ranges, South Australia; black dotted line marks edge of diapir, which has randomly oriented stringers of carbonates, siliciclastics, and volcanics in its upper half and a large stringer with recumbent isoclinal folds in the lower half (adapted from Hearon et al., 2015). (b) Photo of exposed Mt. Sedom diapir in Israel showing vertical beds of halite with bedding-parallel trains of boudins of eroded-out marls showing clockwise rotation and the indicated sense of shear (see Alsop et al., 2015). (c) Cross section of a diapir in northern Germany constrained by subsurface mine data, showing boudins of anhydrite (purple) entrained in complexly folded halite
(adapted from Jackson and Hudec, 2017, itself after Hofrichter, 1980). (d) Photograph of tunnel wall in the Wieliczka salt mine, southern Poland, showing anhydrite-mudstone boudins (medium grey) thrusted and folded within halite matrix (see Krzywiec et al., 2017). (e) 3-D depth-migrated seismic profile (vertical exaggeration 2:1) from the Santos Basin, Brazil, with complex internal deformation shown by layered halite, anhydrite, and bittern salts (data courtesy of PGS Investigação Petrolifera Limitada).

**Figure 3.** (a) Stress-strain rate diagram at a temperature of 60º C illustrating the most likely conditions for evaporite deformation
addressed in this paper. The grey lines represent different halite viscosities, the red line is an extrapolation of laboratory data for dislocation creep of halite, the blue lines are the flow lines for pressure-solution creep for different grain sizes. The intersections of the blue lines with the red line represent steady-state flow where the grain size is in equilibrium with the differential stress. Thus the grey ellipse is an estimate of the stress and strain-rate conditions for the flow of halite in the subsurface, constrained by measurement of the differential stress by subgrain-size piezometry. The purple ellipse is an estimate of the conditions for anhydrite deformation during layer-
parallel shortening based on analysis of fold shapes, and the pink ellipse is an estimate of the condition for bittern salts. (b) Schematic diagram comparing the strength of halite and non-evaporite rocks as a function of depth in extension and contraction (adopted from Weijermars et al., 1993, and Jackson and Hudec, 2017). The purple line is for wet halite, which has a constant, very low strength (<1 MPa); the colored fields are for non-evaporite rocks in extension (blue) and contraction (green), which range from dolostones (very strong) to highly overpressured mudstones (very weak).

**Figure 4.** Conceptual diagrams of layer-parallel extension: (a) undeformed salt layer (of area $A_0$) with two competent internal beds (yellow); (b) boudinage of competent beds into stringers during thin-skinned extension, assuming the primary décollement is near the base of salt; (c) boudinage during thick-skinned extension, concentrated in areas of drape folds over presalt faults.

**Figure 5.** Examples of extension of salt with internal competent beds: (a) simplified results of analog model in which the polymer layer was thinned and attenuated during thin-skinned extension – the black dashed line shows the original top polymer, and the yellow and blue





layers are stronger intra-polymer layers that separated into boudins, with more extension at the deeper, blue level (adapted from Cartwright et al., 2012; model by T. Dooley); (b) legend to colors used in (a); (c) 3-D depth-migrated image of a large salt roller from the Santos Basin (vertical exaggeration 1:1), with very little of the internal layering seen in Fig. 2e visible here due to more disruption during extension (data courtesy of PGS Investigação Petrolifera Limitada); (d) 3-D depth-migrated seismic profile (vertical exaggeration 2:1)
from the Friesland Platform of the onshore Netherlands showing the Zechstein salt (between cyan horizons) with boudins of the Z3 stringer (top of stringer in yellow) generated during thick-skinned extension (adapted from Strozyk et al., 2014).

**Figure 6.** Conceptual diagrams of plane-strain layer-parallel shortening: (a) undeformed salt layer with two competent internal beds (yellow), with different thicknesses and of length $\ell_0$, that divide the halite into three layers with equal area $A_0$; (b) shortening of 14% generates folded overburden (box fold for simplification), with thickening of salt in core of anticline and thinning beneath synclines;
maintaining halite areas, with no breaking or folding of the strong beds, results in apparent decrease in line length of the competent beds; (c) more realistic scenario of buckle folds in competent beds, with the amount of internal shortening increasing downward to maintain bed length and a longer wavelength in the thicker strong layer.

**Figure 7.** Examples of layer-parallel shortening of salt with internal competent beds: (a) simplified results of analog model of shortening detached on polymer layer with a thin interbed of competent sand (yellow) located at the middle level of the polymer – the black dashed
line shows the original top polymer, and the intra-polymer layer forms tight to isoclinal folds that have been partially disrupted (adapted from Pla et al., 2019); (b) simplified results of analog model showing shortened and inflated polymer (this is the other half, but reversed, of the model shown in Fig. 4a) – the black dashed line shows the original top polymer, and the yellow and blue layers are stronger intra-polymer layers, with more contraction at the deeper, blue level (adapted from Cartwright et al., 2012; model by T. Dooley); (c) legend to colors used in (a) and (b); (d) 3-D depth-migrated seismic profile from the Santos Basin (vertical exaggeration 1:1) with a gently folded
LES (between the cyan horizons) and cover – the deep (blue) intrasalt level is shortened more than the shallow (green) level, but this is not due to progressive deformation during evaporite deposition (modified from Fiduk and Rowan, 2012); (e) 3-D depth-migrated seismic profile (vertical exaggeration 2:1) from the Levant Basin showing the mostly tabular Messinian evaporites (LES, between the cyan horizons) with only minor folding of the cover – the shallower MC2 intrasalt sequence (see Fig. 1e) is shortened more than the MC1 level (modified from Feng et al., 2016).

**Figure 8.** Different possible shear profiles during shortening (inspired by Cartwright et al., 2012), with cover deformation not shown for the sake of simplicity: (a) Couette flow (simple shear of entire salt layer), resulting in upward increasing shortening magnitudes for intrasalt competent beds; (b) intrasalt basal shear, with competent beds shortened equally; (c) combined inverse Couette flow and basal shear, with intrasalt shortening decreasing upward; (d) asymmetric Poiseiulle flow, with maximum shortening of intrasalt beds near the center of the salt layer.

**Figure 9.** Possible changes in intrasalt shear profile during progressive shortening and growth of contractional anticlines: (a) incipient shortening accommodated by Couette flow; (b) as synclines sink and zone of shear becomes more concentrated, the Couette flow gets modified; (c) as the salt beneath the synclines thins, intrasalt basal shear becomes increasingly dominant. Complex intrasalt deformation in the anticlinal cores is not shown.

**Figure 10.** Conceptual diagrams of salt inflation and deflation due to the application of a differential sedimentary load (not shown), with
two-dimensional flow of salt only within the model area (salt to right and left of model flows into adjacent salt pillows): (a) undeformed salt layer with two competent internal beds of differing thickness; (b) incipient salt movement into salt pillow, with boudinage of strong layers beneath depocenters and folding within pillows; (c) further deformation of intrasalt layers due to increasing deflation/inflation, with greater amounts of strain at shallower levels if there is insignificant mixing of halite above and below each competent bed.

**Figure 11.** Examples of differential loading of salt with internal competent beds (yellow): (a) simplified sequential results of finite-
element model with subsidence on both ends and inflation into a central salt pillow – the early deformation of the strong layer (top) is dominated by boudinage, but further deformation leads to folding and overlap of the boudins in the pillow (adapted from Raich, 2017); (b) simplified sequential results of numerical model with progradational loading (indicated by black dashed lines) and consequent inflation – the strong layer is pulled almost completely apart beneath the depocenter and is first folded and then thrusted and disrupted in the inflated body (adapted from Albertz and Ings, 2012); (c) legend to colors used in (a) and (b); (d) 3-D depth-migrated seismic profile (vertical
exaggeration 2:1) from the Groningen High of the onshore Netherlands showing a gentle salt pillow of the Zechstein salt (between cyan horizons) between two Triassic depocenters – the Z3 stringer (yellow) is mostly continuous and slightly folded within the pillow but more broken up beneath the depocenters (adapted from Strozyk et al., 2014).

**Figure 12.** Schematic diagram of three-dimensional stress (red arrows) and associated deformation of an intrasalt competent layer (yellow) in a circular to slightly elliptical salt pillow. As salt flows from beneath the depocenters into the pillow (blue arrows), radial extension and



contraction (boudinage and folds) are accompanied by constrictional folding due to convergent flow into the pillow. Drafted from original sketch by F. Peel.

**Figure 13.** Conceptual diagrams of passive diapirism: (a) salt layer of area $A_0$ and with two strong layers (yellow) of length $\ell_0$ (top) is source for salt flowing into diapir with width $w$ and height $h$ (bottom) in plane-strain, constant-area scenario – flow into and up diapir

(blue arrows) results in lengthening and boudinage of strong layers; (b) graph showing extension $e$ of strong layers as a function of the aspect ratio $a$ and the fetch factor $F$ (see text); (c) Poiseuille flow up diapir, as shown by deflection of imaginary horizontal line (red, left) to its deformed equivalent (right) results in shear-induced rotation of stringers (yellow) to near-vertical orientations in most cases.

**Figure 14.** Examples of passive diapirs formed by LES with internal competent beds: (a) 3-D depth-migrated seismic profile (vertical exaggeration 2:1) from the Waddenzee of the onshore Netherlands showing two diapirs formed by the Zechstein salt (between cyan

horizons) – very little of the Z3 stringer (yellow) is visible within the diapirs or beneath the flanking depocenters (adapted from Strozyk et al., 2014); (b) 3-D depth-migrated seismic profile (vertical exaggeration 1:1) from the Santos Basin with expulsion-rollover structure and passive diapir (data courtesy of PGS Investigação Petrolifera Limitada).

**Figure 15.** Simplified sketch based on observations of 3-D seismic data in the northern Red Sea. The bright package (in purple) locally at the base of the Mansiya salt (in grey) is the Kial Formation, which includes significant proportions of anhydrite (Hughes and Johnson,

2005). The Kial has variable expression: as coherent bedding, as slightly disrupted (due to thrusting), as isolated stringers near the base of salt, or absent. Thus, the base of the mobile, transparent salt (grey) cuts up and down stratigraphically.

**Figure 16.** Analog model and seismic profile from the Santos Basin: (a) simplified results of analog model, where A1 is pure silicone polymer and A2-A4 is a mixture of polymer and stronger, more dense granular interlayers (adapted from Dooley et al., 2015); (b) time-migrated 3-D seismic profile at approximately 1:1 scale, with the analogous A1 and A2-A4 layers (adapted from Jackson et al., 2015) – the

purple line is an extra layer within the A1 unit and is part of the B3 sequence of Fiduk and Rowan, 2012); (c) line-length restoration of the analog model of (a) showing that layers fit back together; (d) line-length restoration of the seismic profile of (b), with an extra 3 km of the purple layer and missing lengths of the shallower layers. Restorations carried out in Move™.

**Figure 17.** Regional 3-D depth-migrated seismic profile from the Santos Basin (vertical exaggeration 3:1), separated into two halves and with salt shown in cyan (data courtesy of PGS Investigação Petrolifera Limitada).

**Figure 18.** The Slochteren and Veendam pillows on the Groningen High, the Netherlands (adapted from Raith et al., 2016): (a) 3-D depth-migrated seismic profile (vertical exaggeration 1:1, location shown by dashed line on map); (b) structure map of the Z3 stringer, with warm colors high, cool colors low, white representing gaps in the interpreted layer (yellow on the seismic), and presalt extensional faults in black.



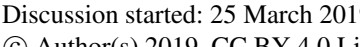



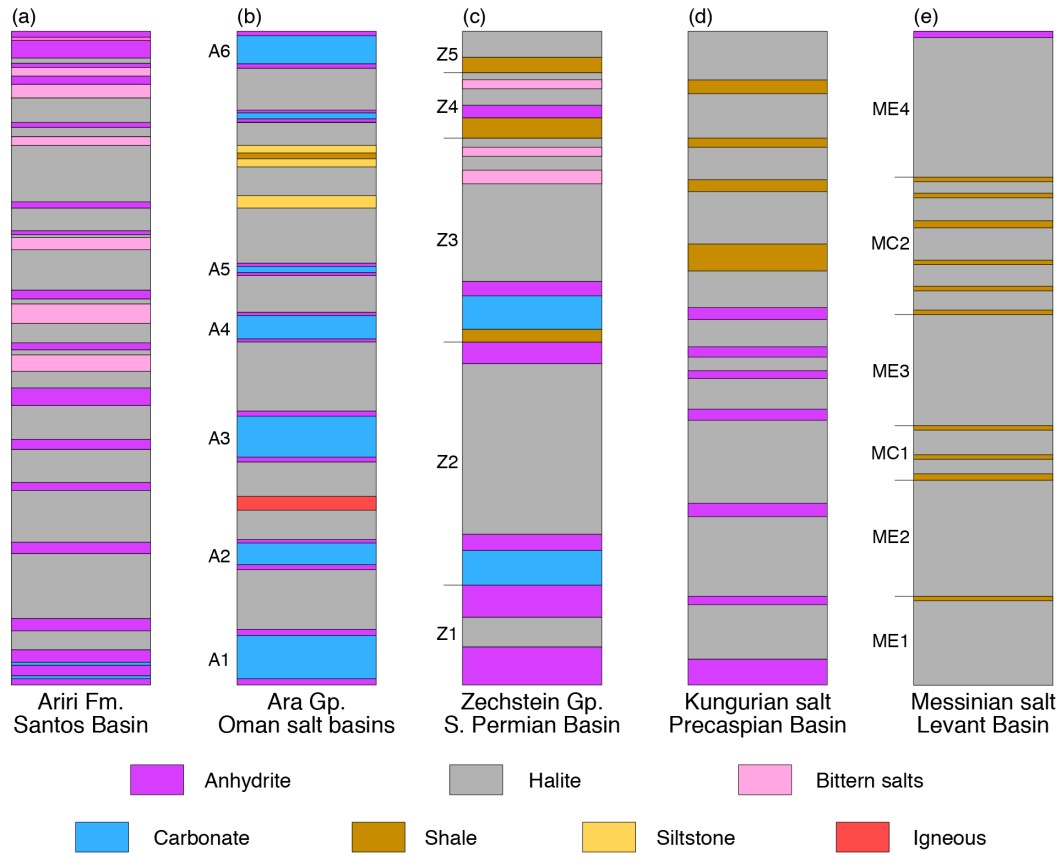

Figure 01



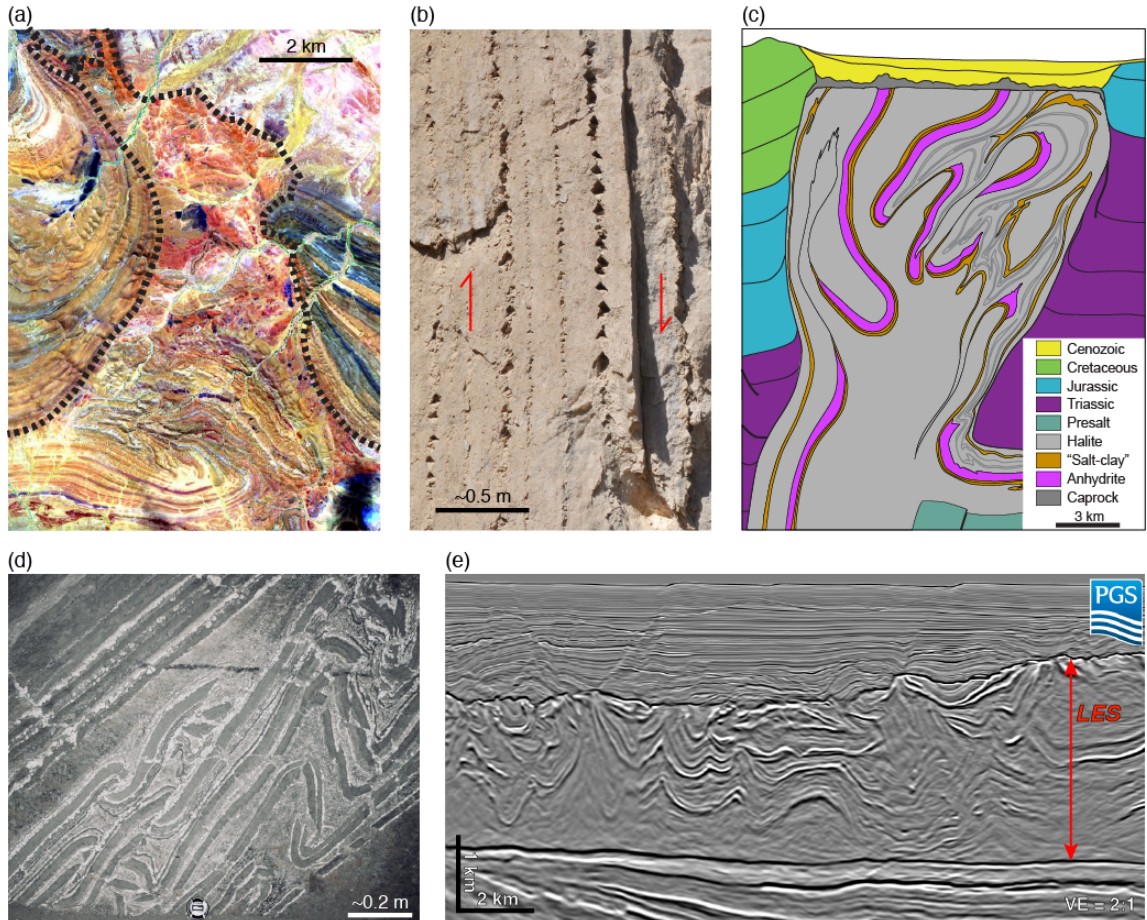

Figure 02



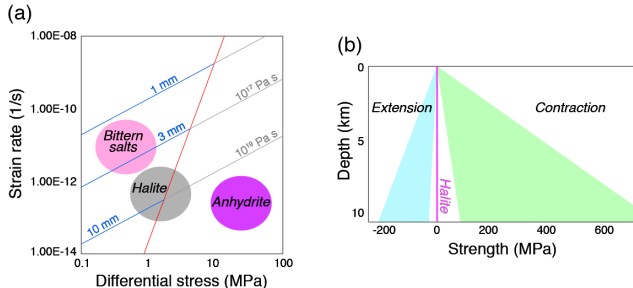

Figure 03



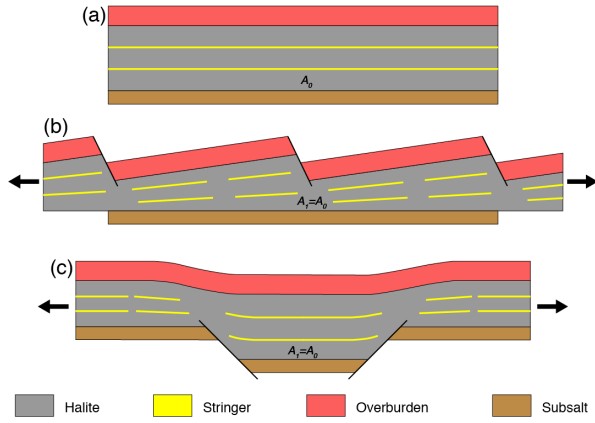

Figure 04



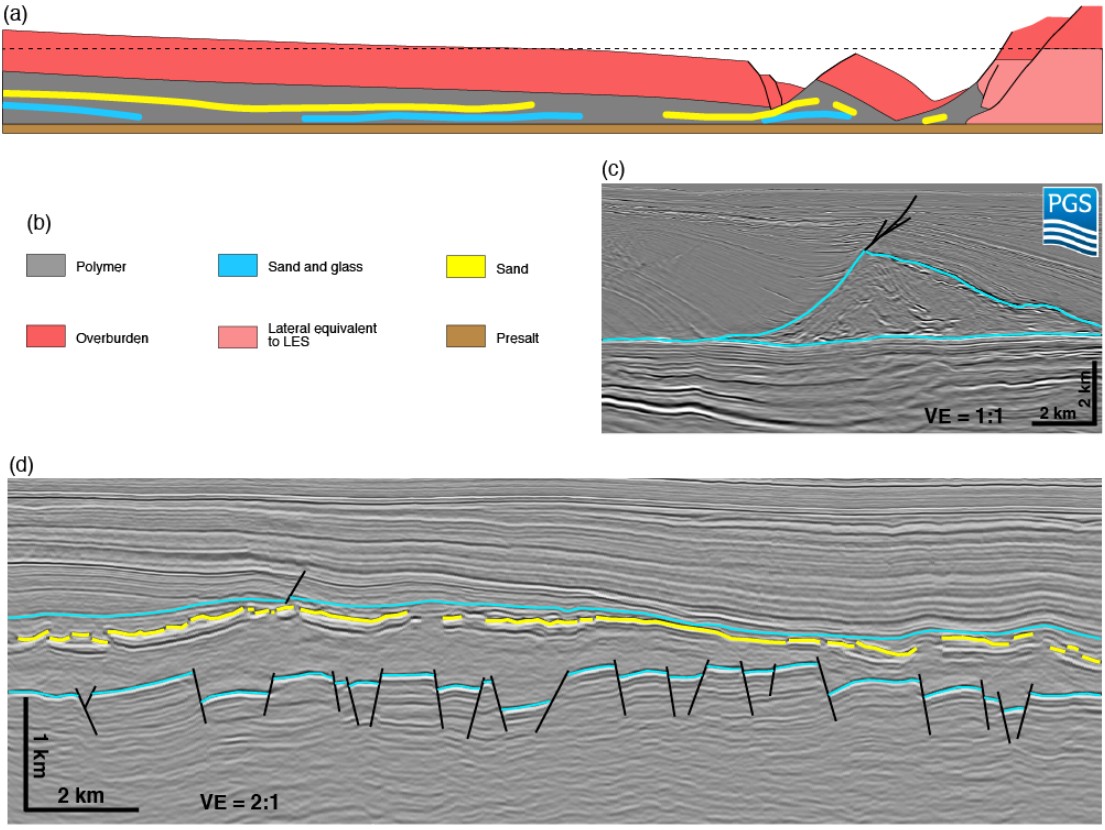

Figure 05





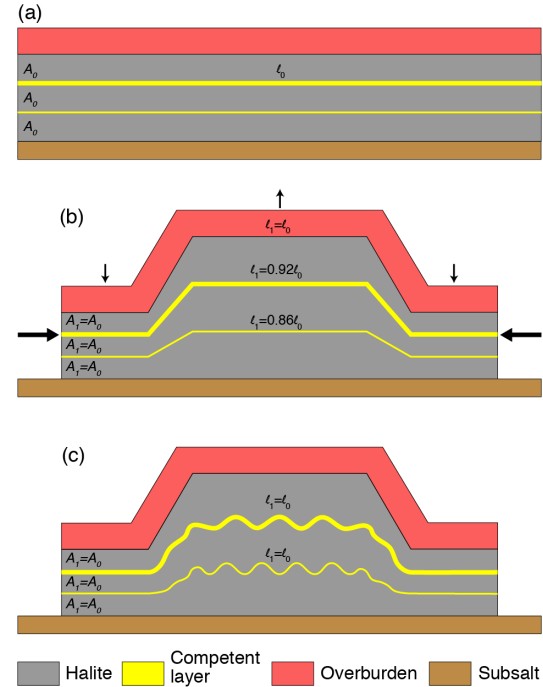

Figure 06



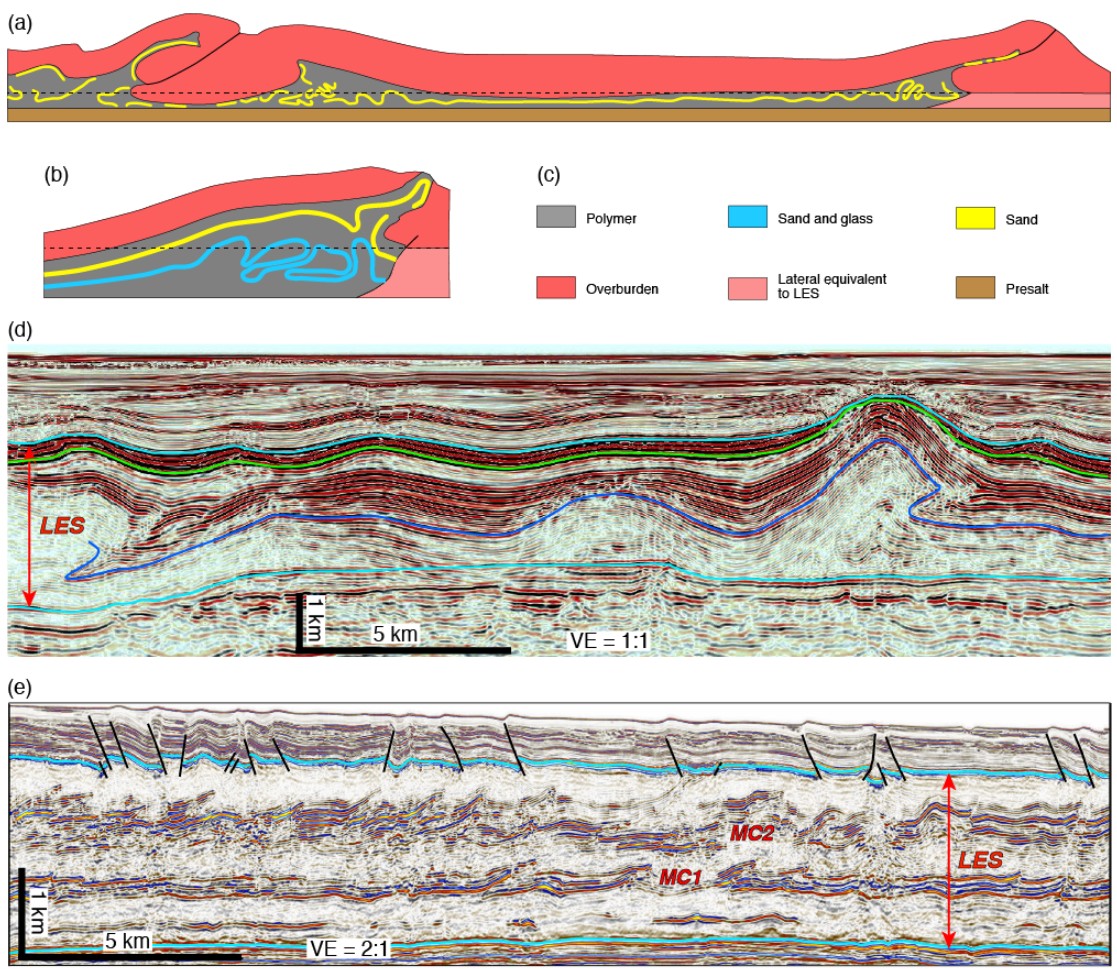

Figure 07



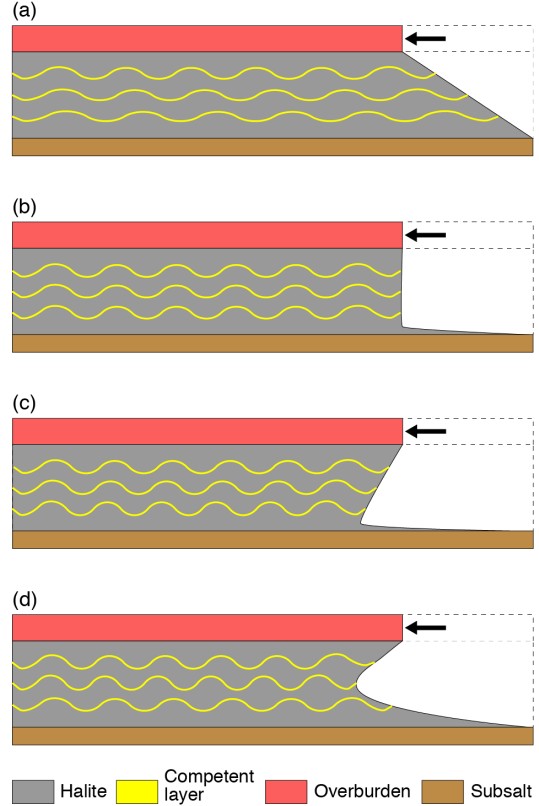

Figure 08




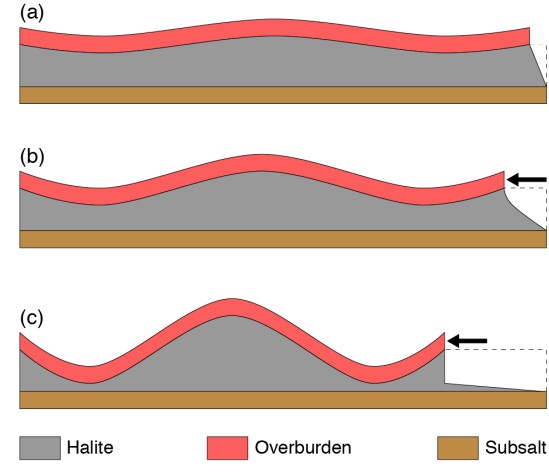

Figure 09





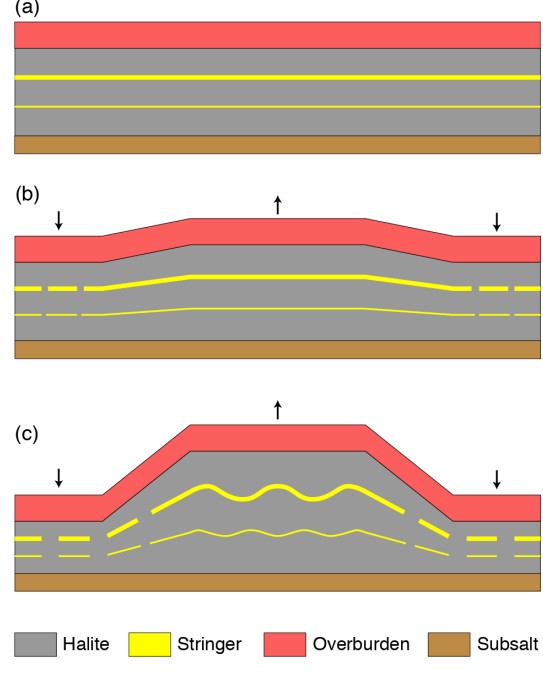

Figure 10





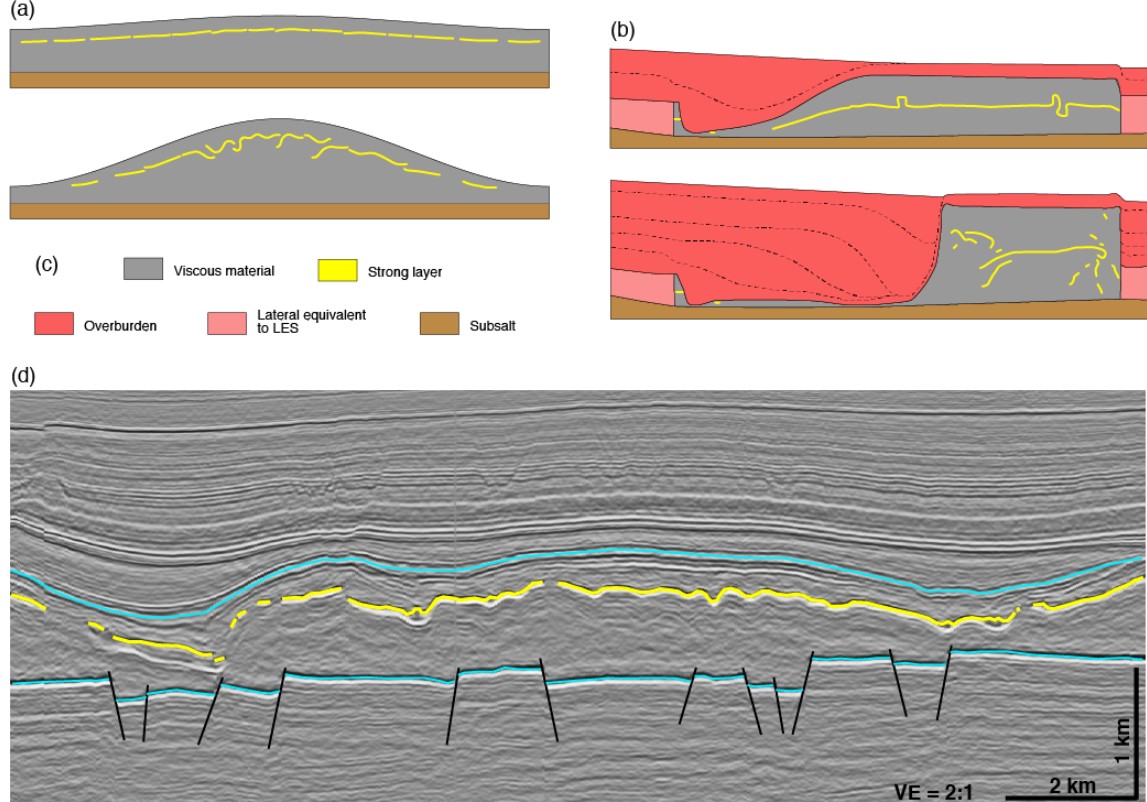

Figure 11



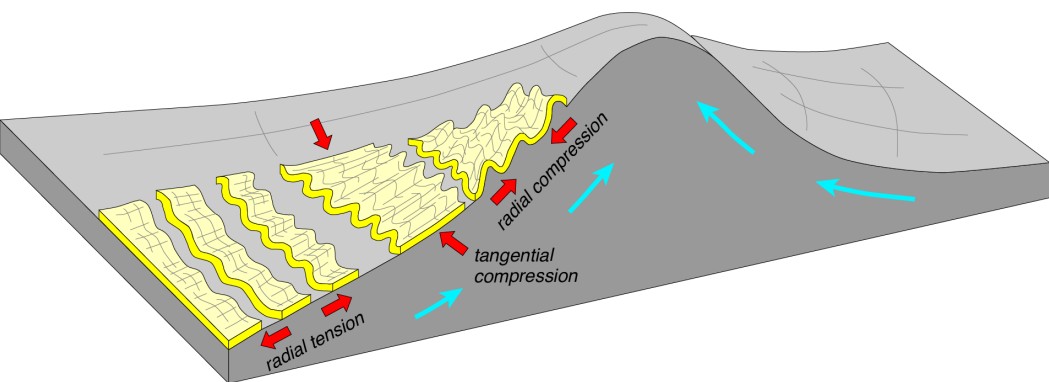

Figure 12





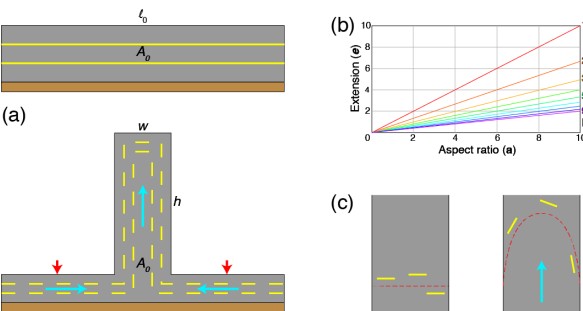

Figure 13



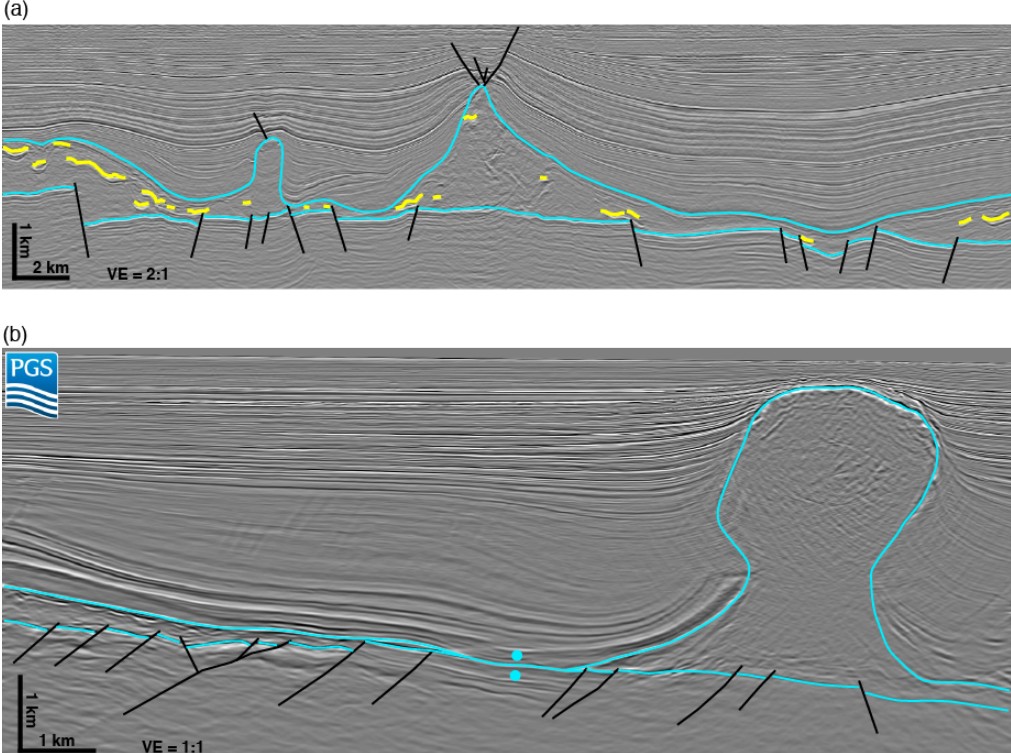

Figure 14

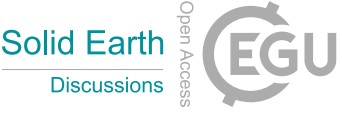

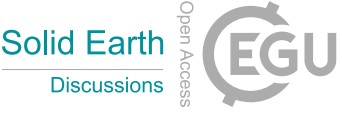

Figure 15



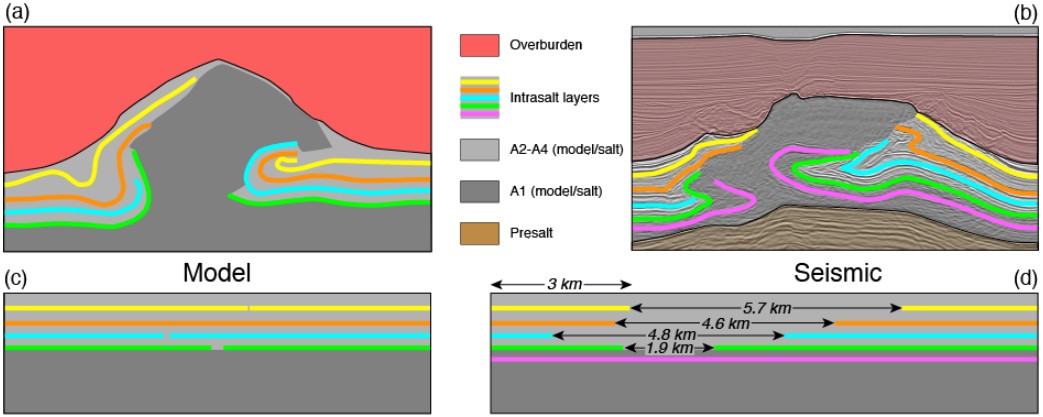

Figure 16



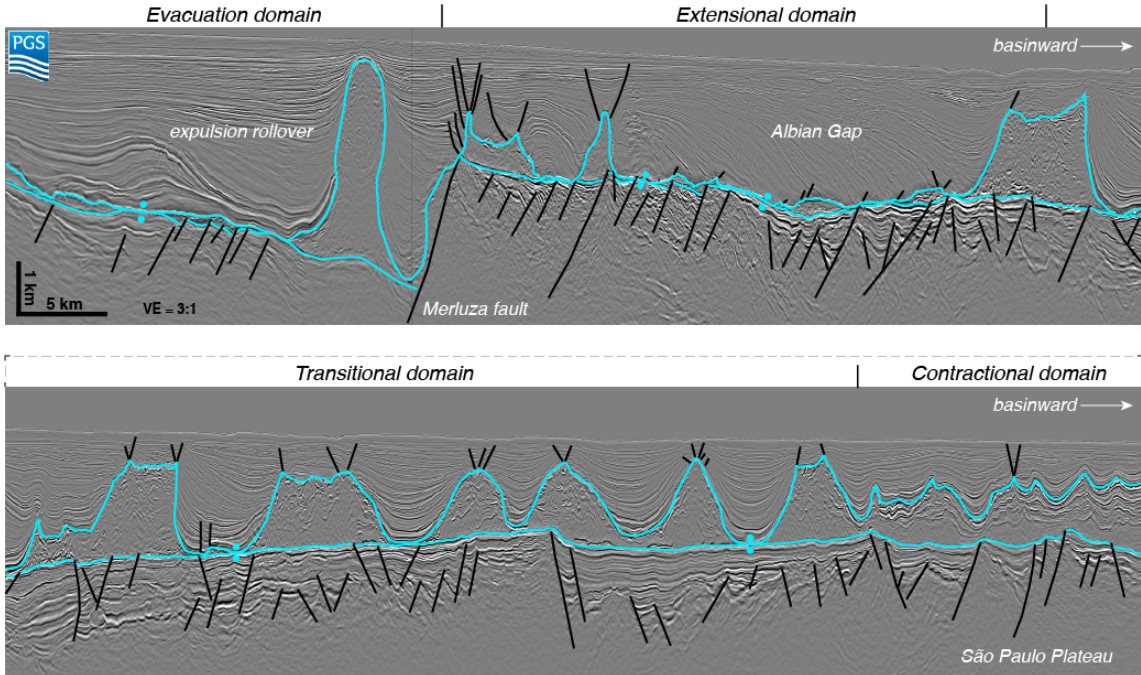

Figure 17



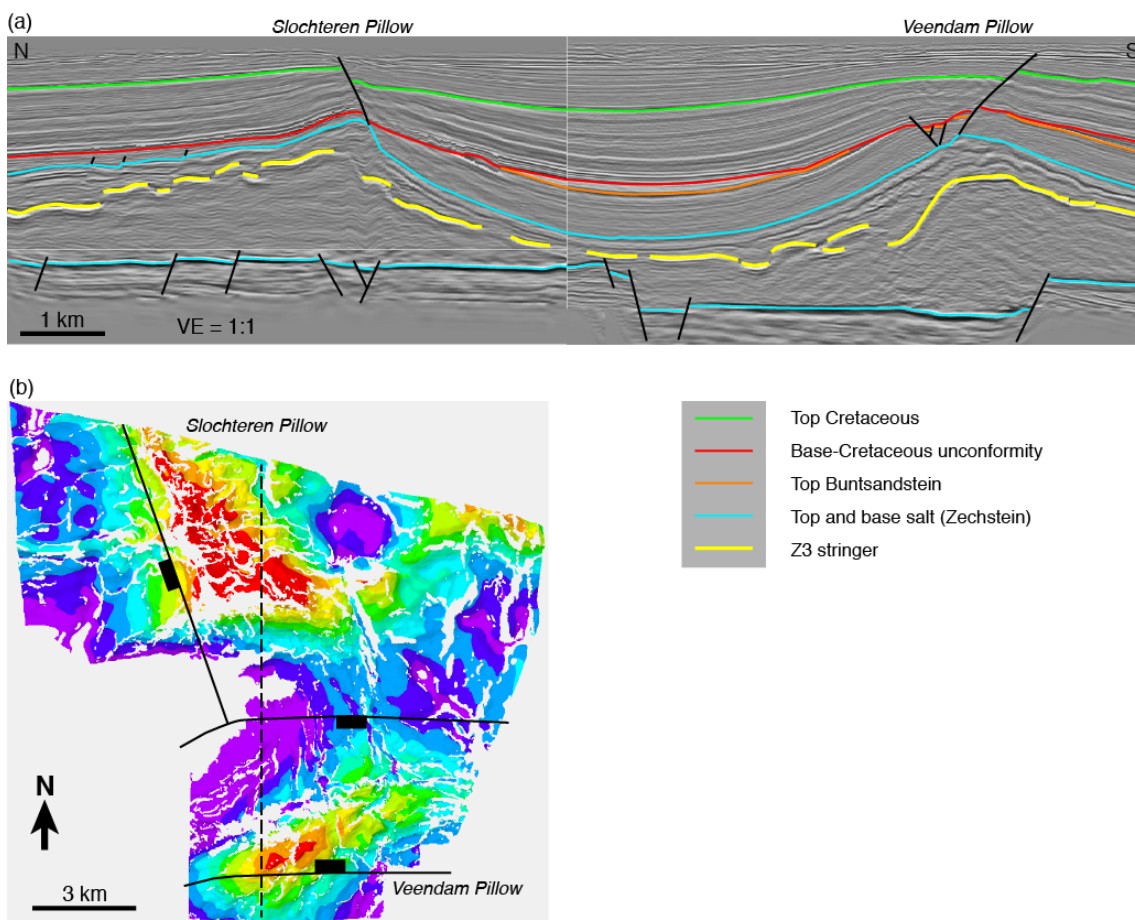

Figure 18