# Peer review of "Deformation of intrasalt competent layers in different modes of salt tectonics"

_Solid Earth, 2019_

## Referee Comment (RC1) · Anonymous Referee #1 · 15 Apr 2019

This paper explores the role of lithological heterogeneities within 'salt' and explores their likely significance. The authors employ a variety of techniques including numerical and analogue modelling together with seismic analysis of salt structures. The authors suggest that competent layers in some settings such as diapirs may be underestimated due to breaking and boudinage of these layers. This is a very well-written and therefore easy to read paper. It should be read by all those who have even just a passing interest in 'salt tectonics' as it clearly explains the main concepts. Although some parts such as 'drilling hazards' seem a little out of place, it does provide a holistic view of the topic. Perhaps the only concerns are that the paper approaches the structural interpretation in terms of 'contraction' (folding etc.) or 'extension' (boudinage etc), and tends to ignore the potential variety of structures that may form in each of these settings via variations

in the actual strain rates. Of course, this is a much more difficult factor to quantify over a given period of time, but it may make for interesting comparisons in different basins in future work. Overall, a well-written and great piece of work – a pleasure to read!

---

## Author Comment (AC1) · 15 Apr 2019

Thank you very much for your kind words. We totally agree that many/most situations have more complex scenarios, with both local spatial and temporal variations, and tried to address this briefly in the Discussion session. But we chose to focus on simple end-member styles as a starting point. We hope that our contribution will stimulate further investigations and conversations on this topic.
* * *

---

## Referee Comment (RC2) · Michael Warsitzka (Referee) · 29 Apr 2019

In the submitted manuscript, Rowan et al. summarize, compare and discuss important findings about intra-salt deformation and the deformation of competent interlayers in layered evaporites. Based on results from published analog experiments, numerical models and seismic data, they explain deformation styles of competent layers during extension, contraction, differential loading and passive diapirism. Then, the influence of various physical parameters, such as layer thickness and strength of the competent layers, on the deformation style is discussed, some specific natural case studies are described and, finally, consequences of the occurrence of deformed stringers for drilling wells and interpreting well and seismic data are emphasized.

The manuscript is well structured and excellently written. The given examples and illustrative sketches provide fundamental guidelines for analyzing and interpreting internal structures in deformed evaporitic succession. In the frame of challenging future tasks like storing waste and resources in salt structures, the particular strength of this research is its timeliness. After going through some minor corrections and suggestions, which are listed below, I recommend final publication.

Comments and Suggestions:

Chapter 3.2 Contraction:

Page 7 line 20: Fig. 7e must be: Fig. 7d

Page 7 line 21: Fig. 7d must be: Fig. 7e

Chapter 3.4 Passive diapirs: In my opinion, the conceptual model of stringer deformation in passive diapirs does not cover all aspects of diapir evolution. For many diapirs, the model of purely dissected and extended stringers dragged into the diapir might be valid. However, there are examples of diapirs, in which the internal deformation is rather similar to those of salt pillows driven by differential loading. For instance, the internal structure of the Gorleben diapir (Bornemann et al., 2003) is characterized by a folded anhydrite layer at least in its Northwestern half. This indicates contraction due to salt influx into the roughly 3km high diapir. This diapir was probably initiated by extension, but likely underwent a long phase of passive growth. Another example is shown in Fig. 8.26 in Jackson & Hudec (2017). The relatively thin anhydrite layer is dissected, but also folded and doubled, which also indicates contraction. Richter-Bernburg (1980) (Fig. 23) displays the top of the Haenigsen diapir (roughly 4 km tall) with complex internal folding. Even if no competent layers are involved in the folding, the diapir interior is dominated by contractional structures. Furthermore, Rowan et al. admitted in the discussion on page 17 that "This is not typical of passive diapirs" for diapirs in the Santos Basin characterized by complexly folded strata.

I think, even if a passive diapir has no precursor deformation, such as extension or contraction, it undergoes an early phase in which salt flux leads to contraction and, therefore, folding of interlayers in the root zone of the diapir. According to the concept in chapter 3.4 these contractional structures should be destroyed due to increasing strain during passive growth. However, numerical models by Chemia & Koyi (2008) (e.g. Fig. 3) show that even at large strains, folds in the deeper parts of the diapir remain and are left behind, while only less dissected stringers are dragged upwards. Furthermore, I think pure passively growing diapirs are rare in nature, since there is always a mechanism needed to initiate the subsidence of adjacent minibasins, e.g. thin-skinned extension, buckle folding, prograding sedimentary wedges, etc.. Therefore, I do not fully agree with the too general statement given by Rowan et al. on page 17: "passive diapirs driven by differential loading have disrupted, rotated strong layers". Furthermore, I suggest to avoid the term "tall diapir" or otherwise: could you provide a height to width ratio defining when a diapir is tall? I suggest to rather provide two alternative concepts: (1) passive diapirs dominated by dissected stringers and (2) passive diapirs dominated by contractional folding.

Chapter 4.2.2 Mechanical stratigraphy of multilayers Page 14, line 4: Fig. 17 is mentioned before Fig. 16, so change the order of both figures.

Chapter 4.4.3 Drilling hazards What about hazards due to peaks in shear stresses occurring at boundaries between competent layers and salt layers? This was modelled and illustrated for instance by Weijermars et al., 2014 and would be worth discussing here in a short paragraph. These stress peaks might be very different depending on whether the competent layer is dissected and floats with the salt or still connected and salt flow is stronger above and below the competent layer.

Reference list: For some references doi are provided, but for most of them not. Please make it consistent.

Figure captions: Caption of Figure 11a: Citation must be "Raith, 2017"

[Figure]

Figures: Figure 18: Please provide a scaled colorbar for Figure 18b.

References

Bornemann, O., Behlau, J., Keller, S., Mingerzahn, G., and Schramm, M.: Standortbeschreibung Gorleben: Teil III – Ergebnisse der Erkundung des Salinars, Bundesanstalt für Geowissenschaften und Rohstoffe, Hanover, Germany, 2003.

Chemia, Z., & Koyi, H. (2008). The control of salt supply on entrainment of an anhydrite layer within a salt diapir. Journal of Structural Geology, 30(9), 1192-1200.

Jackson, M. P. A., and Hudec, M. R.: Salt Tectonics – Principles and Practice, Cambridge University Press, Cambridge, UK, 2017.

Richter-Bernburg, G.: Salt tectonics, interior structures of salt bodies, B. Cent. Rech. Expl., 4, 373-393, 1980.

Weijermars, R., Jackson, M. P. A., & Dooley, T. P. (2014). Quantifying drag on wellbore casings in moving salt sheets. Geophysical Journal International, 198(2), 965-977.

———————————————————————

---

## Author Comment (AC2) · 30 May 2019

Thank you, Michael, for your generous and helpful comments. Below, we respond to the main issues you raise.

We agree that pure passive diapirs, without any precursor extension, contraction, or pillow formation, are relatively rare. They do exist, however, for example the Witchelina diapir in South Australia, where the very first (and subsequent) overburden strata onlapped the growing diapir (paper in review by E. Gannaway-Dalton). This situation is likely in settings with very shallow-water or nonmarine deposition, so that differential loading immediately above the salt creates exposed salt highs and thus truncating (diapiric) relationships between the salt and the flanking strata. In any case, we will state

[Figure]

even more explicitly in the final version of the paper that we are examining end-member scenarios.

But let's address the diapirs in Northern Germany that you cite. You write that these diapirs, which probably have the best-mapped (and published) internal geometries in the world, are dominated by folds. We include an example in our Fig. 2c, which is actually the one you cite as Fig. 8.26 in Jackson and Hudec (2017). These folds are expressed by the different layers of the well-known Zechstein stratigraphy, which of course is dominated by halite, and include both map-view (curtain) folds and cross-sectional isoclinal folds (both with steep and curvilinear axial traces). The primary reasons for such folding are that (1) large areas of salt are typically moving in a convergent manner into the smaller area of a diapir, leading to constrictional strain, and (2) that folding is induced by differential flow and shear between the inner and outer parts of the rising diapir.

So yes, strong layers such as anhydrite, carbonates, and siliciclastics are carried and folded passively within the ductile halite (and bittern salts). However, the same maps and cross sections from Germany show that these strong layers are disrupted into boudins within these folds. In volume-constant strain, there cannot be just shortening, and at least one principal strain is lengthening. So we would respectfully disagree: boudins of strong layers are just as common in these diapirs as folds. Rather than considering two alternative styles (dissected stringers vs contractional folding), we argue that both features are part of the same deformation process. In our opinion, the impression that many passive diapirs are characterized by folding and contraction is created by a common focus on halite and other ductile layers. Our focus, however, is on the strong layers, and we would thus argue that the observations that you cite are actually compatible with our simple models. But we will do a better job in the revised version of making this distinction – between the large-scale halite-dominated folding and the brittle rupturing of strong layers – more clear.

As for the models by Chemia and Koyi, we argue that the material properties used in their numerical models are not appropriate and thus do not create ruptured boudins.
They use a Newtonian-viscous rheology for the salt, which is better modeled as a power-law fluid. More importantly, they applied a power-law (i.e., viscous) rheology to the anhydrite, stating that "the brittle rheology of anhydrite cannot be achieved in numerical models." Thus, although the flow of the modeled anhydrite is appropriate for model folding, it does not allow the brittle rupture process that is well documented in anhydrite in nature (see Abe et al., 2013).

Finally, on the topic of drilling hazards, we agree that shear stresses between rock salt and competent layers can be significant. However, those calculated in Weijermars et al. (2014) are unrealistically high and to our knowledge cannot occur in the subsurface. Having said that, stresses in anhydrite stringers caused by salt flow or gravitational sinking are important but little studied. How there stresses affect possible open fractures in (overpressured) anhydrite is not well known but could well be important.

Again, thank you for your thoughtful comments. They will help us refine our message and minimize any possible confusion.

---

## Author Response (AR2)

**Final response**

**Anonymous Referee #1**

1. Comment – section on drilling hazards seems a little out of place, but does provide a holistic view.
   Response – this section helps show the importance of the topic of the paper.
   Action – none taken.

2. Comment – division into contaction or extension ignores the potential variety of structures in real scenarios.
   Response – we agree and state that we are examining simple end-member styles in order to better understand the fundamental principles and processes, and we already address more complex scenarios in the Discussion.
   Action – we have emphasized even more strongly that these are indeed end-member situations and that this will help us understand the combinations typical of real examples.

**Referee #2 (Michael Warsitzka)**

1. Comment – references to Fig. 7d and 7e need to be switched in the text.
   Response – agreed.
   Action – fixed, but other changes mean that it's now 7c aqnd 7d.

2. Comment – many passive diapirs are dominated by contractional folding, not extended stringers as we show. Suggests providing two alternative models: passive diapirs dominated by boudins of strong layers, and passive diapirs dominated by contractional folding.
   Response – we responded to this in detail in the "Author Response". To summarize, we argue that both styles actually coexist in passive diapirs. Yes, the ductile halite and bittern salts are complexly folded, but strong layers are indeed ruptured into boudins within these larger-scale folds. Our focus in this paper is on the strong layers, thus our models are compatible with the observations cited by the Referee.
   Action – we have made these points more explicit in the text, distinguishing more clearly between the deformation of the weak and strong layers.

3. Comment – pure passive diapirs are rare in nature.
   Response – we agree, but they do exist. Moreover, our paper is focused on simple end-member styles, as explained in our response above to comment #2 by Referee #1.
   Action – again, we have tried to make these points more clearly in the revised text, emphasizing both the simple end members and the commonly more complex real-world examples.

4. Comment – avoid the term "tall diapir".
   Response – agreed that this is poorly defined.
   Action – we have removed the term when used in a general sense, but kept it when describing a specific diapir in our figures or when contrasting between tall and short.

5. Comment – reference to Fig 17 is made prior to that for Fig 16, so order should be changed.
   Response – agreed that the order in the text is not correct.
   Action – kept the order of the figures and removed the early mention of Fig 17.

6. Comment – would be worth mentioning modeling by Weijermars et al. (2014).
   Response – we think the modeled stresses are too high, but agree that it is worth mentioning.
   Action – we added two sentences addressing this issue and citing this work.

7. Comment – doi numbers are missing for many references.
   Response – agreed.
   Action – fixed.

8. Comment – name misspelled in caption of Fig 11a.
   Response – agreed.
   Action – fixed.

9. Comment – add colorbar to Fig 18b.
   Response – agreed.
   Action – done.